# The p97-UBXD8 complex regulates ER-Mitochondria contact sites by altering membrane lipid saturation and composition

Rakesh Ganji [1], Joao A. Paulo [2], Yuecheng Xi[3], Ian Kline[3], Jiang Zhu[4,7], Christoph S. Clemen [5,6], Conrad C. Weihl[4], John G. Purdy [3], Steve P. Gygi [2] & Malavika Raman [1] ✉

The intimate association between the endoplasmic reticulum (ER) and mitochondrial membranes at ER-Mitochondria contact sites (ERMCS) is a platform for critical cellular processes, particularly lipid synthesis. How contacts are remodeled and the impact of altered contacts on lipid metabolism remains poorly understood. We show that the p97 AAA-ATPase and its adaptor ubiquitin-X domain adaptor 8 (UBXD8) regulate ERMCS. The p97-UBXD8 complex localizes to contacts and its loss increases contacts in a manner that is dependent on p97 catalytic activity. Quantitative proteomics and lipidomics of ERMCS demonstrates alterations in proteins regulating lipid metabolism and a significant change in membrane lipid saturation upon UBXD8 deletion. Loss of p97-UBXD8 increased membrane lipid saturation via SREBP1 and the lipid desaturase SCD1. Aberrant contacts can be rescued by unsaturated fatty acids or overexpression of SCD1. We find that the SREBP1-SCD1 pathway is negatively impacted in the brains of mice with p97 mutations that cause neurodegeneration. We propose that contacts are exquisitely sensitive to alterations to membrane lipid composition and saturation.

Contact sites between the ER and mitochondria allow for compartmentalization of biosynthetic reactions such as lipid synthesis, calcium transport and apoptosis among others[1–3]. ER–Mitochondria contact sites (ERMCS) form when the membrane of these organelles come into close apposition (observed to be between 5 and 100 nm) without fusion[2,4]. ERMCS can form and dissociate in a dynamic manner but can also be stably associated for several minutes. These contacts can be discrete or the organelles can interact extensively wherein an ER tubule can fully envelope a mitochondrion[5]. ERMCS are stabilized by the interaction between tethering proteins that reside in apposing membranes. Several well-characterized tethers have been described that regulate Ca$^{2+}$ transfer and other processes at ERMCS, including

VAMP associated protein B (VAPB)–protein tyrosine phosphatase interacting protein 51 (PTPIP51), mitofusins 1 and 2 (MFN1/2), and inositol triphosphate receptor (IP3R)–voltage-dependent anion channel 1 (VDAC1). Furthermore, enzymes that mediate phospholipid biosynthesis are located on both the ER and mitochondrial membranes and intermediates in phospholipid synthesis are translocated between these two organelle membranes at ERMCS[6]. How lipid biosynthesis and modification at ERMCS impacts the levels of these lipids or membrane properties throughout the cell is presently unknown.

Regulated protein degradation via the ubiquitin–proteasome system is an efficient means to modulate contacts as numerous ubiquitin-reliant protein quality control mechanisms surrounding the

---

[1]Department of Developmental Molecular and Chemical Biology, Tufts University School of Medicine, Boston, MA, USA. [2]Department of Cell Biology, Harvard Medical School, Boston, MA, USA. [3]Department of Immunobiology, BIO5 Institute, University of Arizona College of Medicine, Tucson, AZ, USA. [4]Department of Neurology, Washington University School of Medicine, Saint Louis, MO, USA. [5]Institute of Aerospace Medicine, German Aerospace Center, Cologne, Germany. [6]Center for Physiology and Pathophysiology, Institute of Vegetative Physiology, Medical Faculty, University of Cologne, Cologne, Germany. [7]Present address: Ilumina Inc., San Diego, CA, USA. ✉e-mail: malavika.raman@tufts.edu

ER and mitochondria can be co-opted to modulate the contact site proteome[7,8]. The p97 AAA-ATPase (also known as VCP) is an abundant, evolutionarily conserved, ubiquitin-selective unfoldase that regulates multiple protein quality control pathways surrounding the ER and mitochondria. p97 identifies ubiquitylated substrates through a host of dedicated adapter proteins that serve to recruit p97 to substrates primarily for degradation. ATP hydrolysis by p97 results in extensive conformational changes that generates the force required to unfold the substrate by pulling it through the central pore of the homo-hexamer[9,10]. Thus, diverse p97-adapter complexes can regulate an array of processes ranging from ER-associated degradation (ERAD), mitochondrial homeostasis, and cell cycle control[11–13]. Thus, p97 is ideally positioned to regulate contacts by mediating the extraction and degradation of membrane-embedded tethers or contact resident proteins. Indeed, the degradation of MFN2 by p97 releases mito-chondria from ER to promote mitophagy[14].

Here we show that p97 and its ER-tethered adapter ubiquitin X domain 8 (UBXD8) localize to ERMCS and are required to maintain the normal repertoire of contacts. Furthermore, loss of this complex increases the number of ERMCS. We demonstrate that p97-UBXD8 regulate the degradation of regulator insulin induced gene 1 (INSIG1) to activate the sterol regulatory element-binding protein 1 (SREBP1), which is enriched at ERMCS. Loss of SREBP1 activation leads to decreased transcriptional activation and protein abundance of lipid desaturases such as stearoyl CoA desaturase 1 (SCD1). This in turn causes an increase in lipid saturation and increases membrane order. We find that ERMCS isolated from UBXD8 knockout cells are enriched in lipids containing saturated or monounsaturated tails and depleted in lipids containing polyunsaturated tails. Quantitative proteomics also indicates altered levels of numerous lipid biosynthetic enzymes at ERMCS in UBXD8 deleted cells. Highlighting the importance of this process, cells express-ing p97 disease mutations that cause multi-system proteinopathy-1 (MSP-1) have an increase in ERMCS. Notably, the activation of SREBP1 and the protein levels of SCD1 are depleted in mouse models of MSP-1. Taken together, our studies find that p97-UBXD8 alter ERMCS by reg-ulating membrane lipid saturation through the SREBP1-SCD1 pathway.

## Results

### UBXD8 is enriched at ERMCS and is required for recruitment of p97

We isolated ERMCS (also known as mitochondria-associated mem-branes (MAM)) from HEK293T cells and probed for p97 and select adapter proteins. Percoll gradient centrifugation of crude mitochon-dria releases associated ER membranes allowing for the purification of these contacts[15]. We used fatty acid coenzyme A ligase 4 (FACL4)[16], VAPB[17] and sigma-1 receptor (SIGMA1R)[18] which are known ERMCS resident proteins, as well as markers for mitochondria (TOMM20) and ER (SEC61β), and found that the ER-localized p97 adapters UBXD8 and UBXD2 (but not cytosolic UBXN1) are enriched at contacts (Fig. 1a, b and Supplementary Fig. 1a). We note that UBXD8 is also found in mitochondrial fractions consistent with recent reports (Fig. 1a)[19,20]. We additionally imaged Cos-7 cells transiently transfected with mito-BFP, SEC61β and mCherry-UBXD8 to observe UBXD8 localization at con-tacts and found that mCherry-UBXD8 was enriched at ERMCS relative to the ER marker SEC61β (Fig. 1c, d). We noted that p97 was present but not obviously enriched in MAM fractions (Fig. 1a and Supplementary Fig. 1a, c, d), therefore we asked whether UBXD8 was required for the localization of p97 to ERMCS. We generated UBXD8 knockout (KO) HEK293T cell lines using CRISPR gene editing (Supplementary Fig. 1b) and found that the abundance of p97 in MAM fractions was decreased in the absence of UBXD8 (Supplementary Fig. 1c, d).

### Loss of p97-UBXD8 increases ERMCS

To determine the role of p97 complexes at contacts, we used a split luciferase reporter wherein the N-terminal fragment of luciferase is targeted to mitochondria and the C-terminal fragment to the ER using established targeting sequences[21,22]. Functional luciferase activity is reconstituted when the two organelles establish close range contacts and luciferase activity can be measured using the live-cell substrate, Enduren. We verified the functionality of the reporter system in wild-type cells by over-expressing receptor accessory protein 1 (REEP1)[21], as well as established ERMCS tethers VAPB and PTPIP51[17,23] and found an increase in ERMCS as expected (Supplementary Fig. 2a, b). To deter-mine what role if any p97-adapter complexes may have at ERMCS, cells were transfected with split luciferase cDNAs and siRNAs and lumi-nescence was measured. Loss of p97 and UBXD8 resulted in an increase in ERMCS (Fig. 2a, b). The increase in luminescence was not due to the inappropriate stabilization of the individual split luciferase reporter constructs in p97-UBXD8 siRNA depleted or UBXD8 KO cells as verified by immunoblotting (Supplementary Fig. 2c, d). In contrast, depletion of UBXD2 which was enriched at ERMCS had no impact on luminescence (Fig. 2c). Furthermore, depletion of another p97 adapter UBXD7 as well as the UFD1-NPL4 hetero-dimeric complex that parti-cipates in many p97-dependent processes had no impact on ERMCS (Fig. 2c).

To verify the specificity of the increased ERMCS phenotype, we expressed wildtype p97 or UBXD8 siRNA-resistant cDNAs and were able to rescue the contact defect (Fig. 2a, b and Supplementary Fig. 2e). However, individual ATP catalytic site mutants in p97 were unable to rescue suggesting that catalytic activity was important for this process (Fig. 2a). We next assessed the role of individual domains in UBXD8, by expressing point mutants in the ubiquitin associated (UBA) or ubiquitin-X (UBX) domains that serve to bind ubiquitin and p97 respectively, as well as deletion of the UAS domain that has been reported to mediate the oligomerization of UBXD8[24]. Loss of these functional domains in UBXD8 also prevented rescue of increased ERMCS (Fig. 2b and Supplementary Fig. 2e). Furthermore, when we over-expressed myc-tagged p97 in UBXD8 depleted cells, we did not observe any change in ERMCS suggesting that p97 and UBXD8 func-tion as a complex (Supplementary Fig 2f).

The p97-UBXD8 complex functions in ERAD to maintain ER quality control[25]. We asked if HMG CoA reductase degradation 1 (HRD1) and GP78, two major E3 ligases that execute ERAD in con-junction with p97 contributed to the ERMCS phenotype. While depletion of HRD1 did not measurably impact contacts, GP78 KO[26] cells had an increase in contacts (Fig. 2d, see below). Loss of p97-UBXD8 function in ERAD can activate the unfolded protein response to alleviate ER stress. To determine whether altered ERMCS observed upon depletion of p97-UBXD8 is a secondary effect of ER stress caused by defective ERAD, we treated cells with ER stressors tunicamycin and thapsigargin which cause protein misfolding in the ER. However, no impact on contacts was observed under these conditions (Supple-mentary Fig. 2g). Furthermore, we probed for the ER chaperone BiP which is induced upon ER stress and found no difference in BiP levels between wildtype and UBXD8 KO cells (Supplementary Fig. 2h). We conclude that the increase in contacts observed in p97-UBXD8 loss of function cells is not a secondary consequence to altered ER protein homeostasis.

The split luciferase assay did not allow us to directly visualize ERMCS, so we performed additional studies to better visualize these contacts upon loss of p97-UBXD8. We transfected cells with previously validated split-GFP based contact site sensor (SPLICS) that consists of two non-fluorescent fragments of GFP fused to targeting sequences to either the outer mitochondrial membrane (OMM-GFP$_{1-10}$) or ER (ER-β$_{11}$)[27] (Supplementary Fig 3a). We used a version of the sensor that reports on close range contacts between the ER and mitochondria (~8–10 nm)[27]. Establishment of close-range contacts between the ER and mitochondria restores GFP fluorescence resulting in the formation of puncta that can be imaged. We found that depletion of UBXD8 or inhibition of p97 catalytic activity with the ATP competitive inhibitor

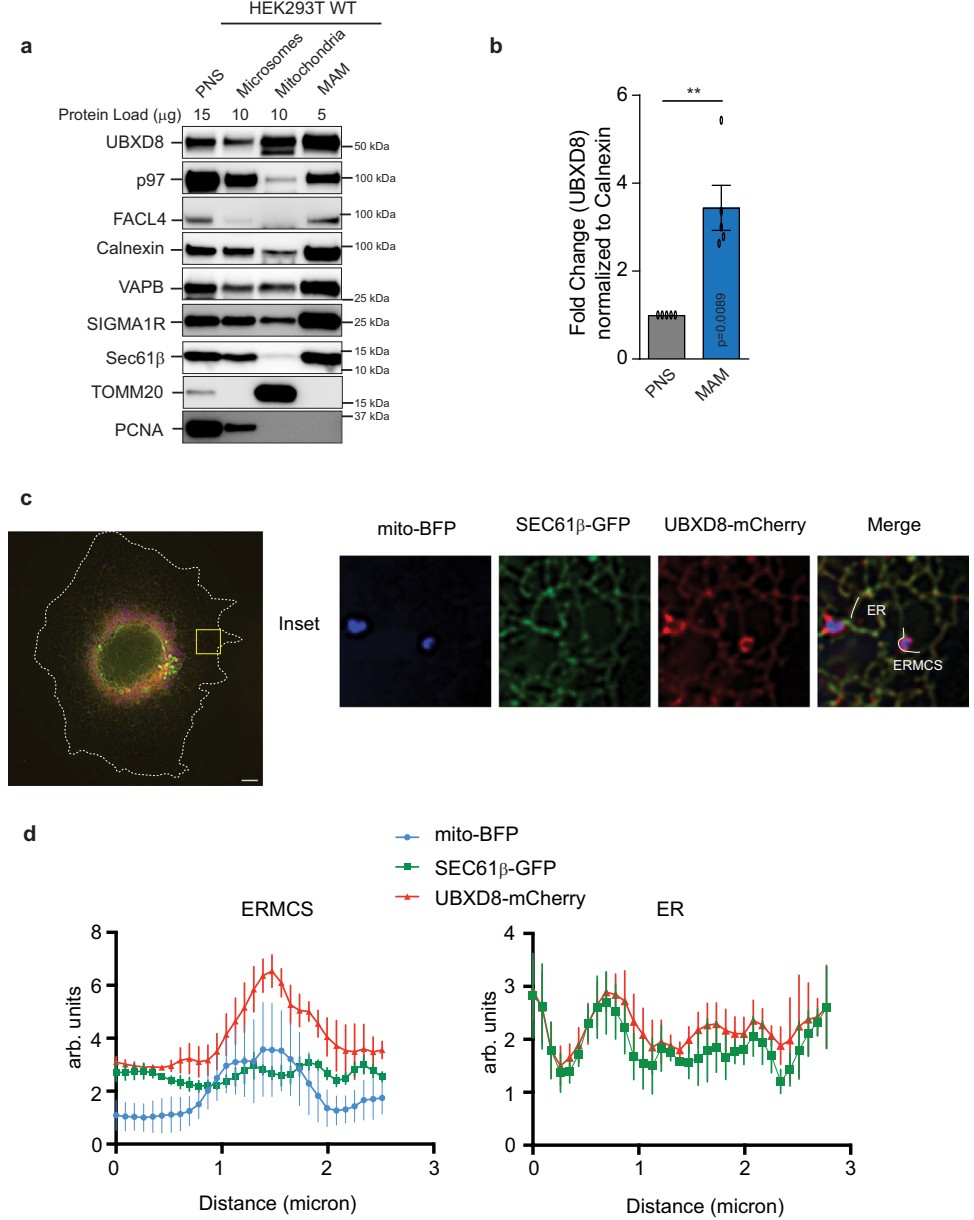

**Fig. 1 | UBXD8 is enriched at ERMCS. a** Immunoblot of the indicated proteins from subcellular fractionation of HEK293T cells. PNS post-nuclear supernatant, MAM mitochondria-associated membrane (*n* > 3 biologically independent samples). **b** Quantification plot showing the enrichment of UBXD8 in MAMs from **a** as normalized to calnexin. (*n* = 5 biologically independent samples). Data are means ± SEM (**\*\*P* < 0.01, two-tailed paired *t* test). **c** Confocal microscopy showing enrichment of UBXD8 at ERMCS using Cos-7 cells transiently transfected with mito-BFP, SEC61β, and mCherry-UBXD8. Insets depict the zoomed in images. Scale bar, 10 μm. **d** Representative line-scan analyses of **c** showing the enrichment of mCherry-UBXD8 at ERMCS relative to SEC61β at ER as determined from *n* = 5 independent cells. Source data are provided as a Source data file.

CB-5083 resulted in an increase in GFP puncta in agreement with our previous observations (Fig. 3a, b). One caveat of the split-GFP system is that once the GFP tether is reconstituted, the contacts cannot effectively dissipate. Hence, we next measured contacts in wildtype and UBXD8 KO cells by measuring the extent of co-localization between the ER and mitochondria by confocal microscopy (Supplementary Fig. 3b, c), and by transmission electron microscopy (TEM) (Fig. 3c). UBXD8 KO cells had a significant increase in ER tubules closely apposed to mitochondria (Fig. 3c, d and Supplementary Fig. 3b, c). No defects in overall mitochondria number or morphology were apparent in UBXD8 KO cells (Fig. 3f). The average length of ER tubules was also unchanged in UBXD8 KO cells (Fig. 3e). Taken together, our findings suggest that the p97-UBXD8 complex has a novel role in regulating ERMCS.

## Quantitative proteomics of ERMCS identifies altered levels of lipid biosynthetic enzymes in UBXD8 deleted cells

To identify pathways altered in UBXD8 KO cells that may contribute to the contact site defect, we isolated ERMCS from triplicate wildtype and UBXD8 KO HEK293T cells by biochemical fractionation and performed multiplexed, quantitative proteomics on the post-nuclear supernatant and MAM fractions using tandem mass tags (TMT) (Fig. 4a, b, Supplementary Fig. 4a–f, and Supplementary Dataset 1)[28,29]. While many known MAM-resident proteins were identified in the PNS sample, they were significantly enriched in the MAM fractions (Fig. 4c). We used two filtering criteria for downstream analysis: |log₂ WT:KO ratio| > 0.65 and −log₁₀ *p* value > 1.5. We quantified a total of 4499 proteins from this study; 102 proteins were enriched, and 112 proteins were depleted in the MAM fraction of UBXD8 KO cells (Fig. 4b and Supplementary

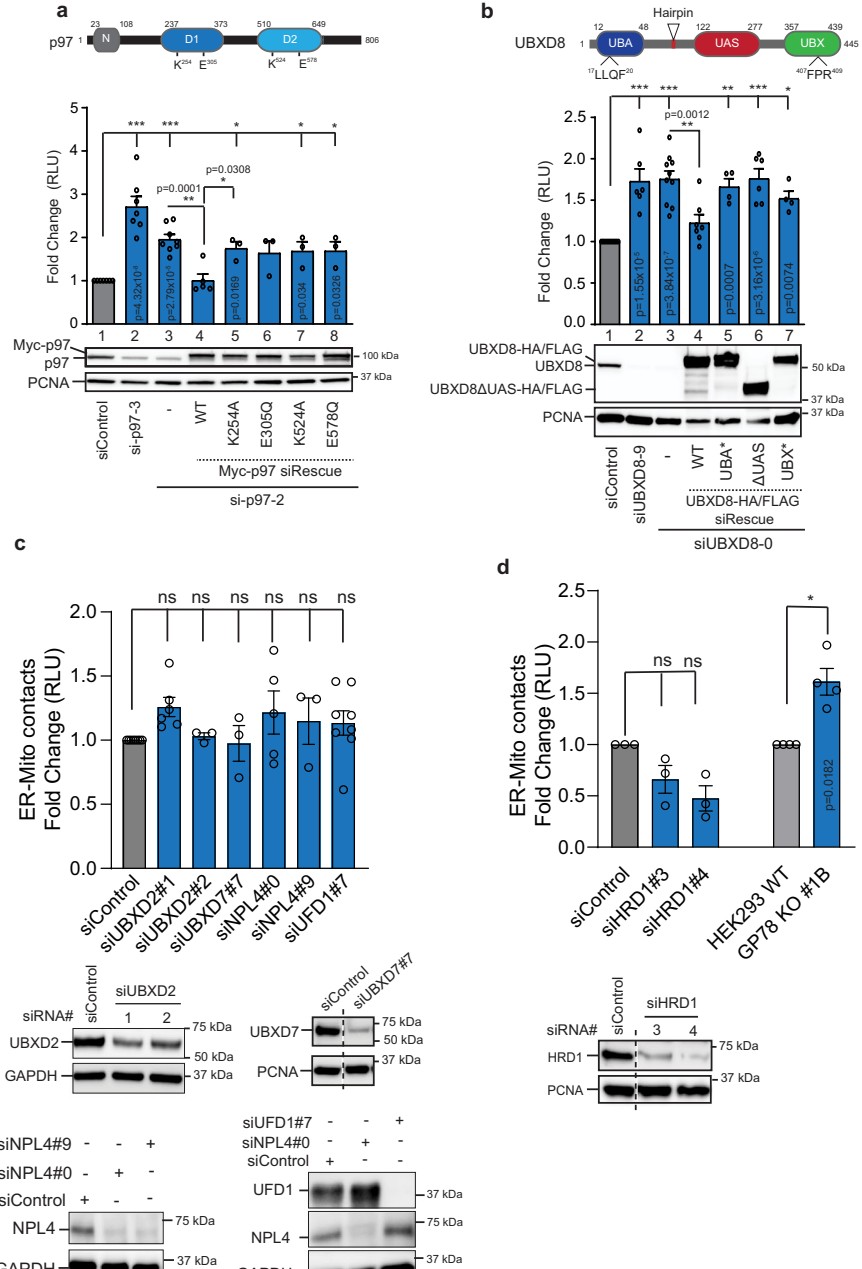

**Fig. 2 | siRNA-mediated repression of p97 and UBXD8 results in increased ERMCS. a** Top: Domain organization of p97 and indicated mutations, Middle: Split luciferase assay to measure contacts in HEK293T cells transfected with siRNAs to p97 and indicated N-Myc siRNA-resistant rescue constructs, RLU relative luminescence unit, Bottom: Immunoblot of indicated proteins ($n = 9, 7, 8, 5, 3, 3, 3,$ and 3 biologically independent samples from left to right, respectively). **b** Top: Domain organization of UBXD8 and indicated mutations, Middle: Split luciferase assay to measure contacts in HEK293T cells transfected with siRNAs to UBXD8 and indicated C-HA/FLAG siRNA-resistant rescue constructs, RLU relative luminescence unit, Bottom: Immunoblot of indicated proteins. UBA ubiquitin associated, UAS upstream activating sequence, UBX ubiquitin X. ($n = 10, 6, 10, 7, 4, 6,$ & 4 biologically independent samples from left to right, respectively). **c** Top: Split luciferase assay to measure contacts in HEK293T cells transfected with the indicated siRNAs

against the p97 adapters. ($n = 8, 6, 3, 3, 5, 3,$ and 8 biologically independent samples from left to right, respectively). Bottom: Immunoblot of HEK293T cells transfected with indicated siRNAs. **d** Top: Split luciferase assay to measure contacts in HEK293T cells transfected with two independent siRNAs against HRD1 or in HEK293 WT or GP78 KO cells. ($n = 3$ biologically independent samples of siControl, siHRD1#3, and siHRD1#4; $n = 4$ biologically independent samples of WT and GP78 KO). Bottom: Immunoblot of HEK293T cells transfected with indicated siRNAs. Data are means ± SEM. *, **, ***$P < 0.05, 0.01, 0.0001$ respectively, ns: not significant, one-way ANOVA with Tukey's multiple comparison test (**a, b, d:** siControl vs siHRD1); one-way ANOVA with Dunnett's multiple comparisons test (**c**); Two-tailed paired $t$ test for **d:** HEK293 WT vs GP78 KO #1B. Source data are provided as a Source data file.

Fig. 4a). Putative contact site proteins identified in previous studies were present in our dataset validating the utility of our approach (Fig. 4b, c and Supplementary Fig. 4b, Supplementary Dataset 1)[21,30–32]. Furthermore, we identified significant enrichment of known p97-UBXD8 substrates such as HMG-CoA reductase (HMGCR)[33,34], and

squalene monooxygenase (SQLE)[35] in UBXD8 KO cells (Fig. 4b and Supplementary Fig. 4g, h). Interestingly, proteins involved in lipid or cholesterol metabolism and lysosome function were also enriched in the UBXD8 KO contact proteome (Fig. 4b, d, e and Supplementary Fig. 4c). In summary, our quantitative proteomic studies of the ERMCS

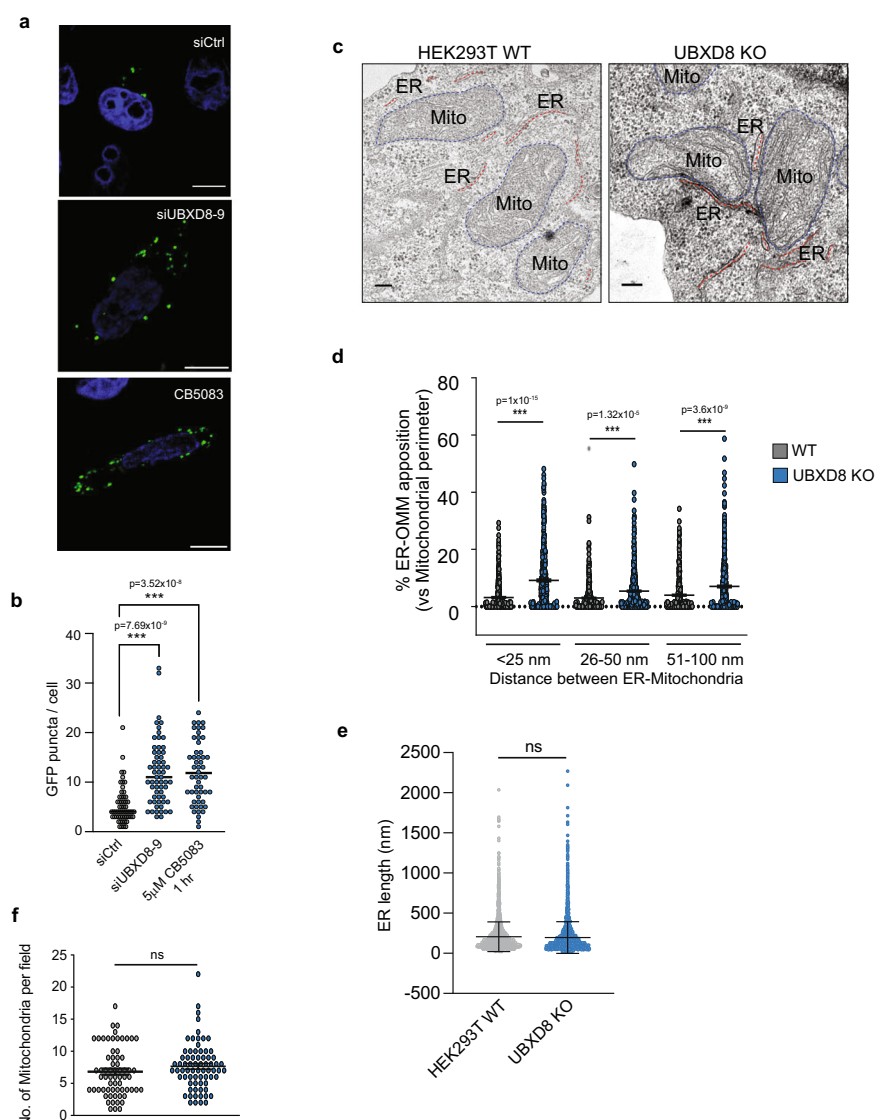

**Fig. 3 | UBXD8 knock out or p97 inhibition results in increased ERMCS.**
**a** Representative microscopy images showing ERMCS as depicted by formation of fluorescent GFP puncta at ERMCS by split-GFP-based contact site sensor (SPLICS) system comprising OMM-GFP$_{1-11}$ and ER-short-GFP$_{\beta11}$[27]. Scale bar, 10 μm. **b** Quantification of GFP puncta per cell as a proxy for ERMCS upon siRNA-mediated repression of UBXD8 or p97 inhibition by small molecule inhibitor, CB5083 (5 μM, 1 h). (Cell numbers used for quantification: siControl = 57 cells; siUBXD8#9 = 61 cells; CB5083 = 52 cells, across $n$ = 3 biologically independent samples). **c** Representative TEM micrographs of wildtype and UBXD8 KO HEK293T cells illustrating contacts between ER (red dotted line) and mitochondria (blue-dotted line). **d** Quantification of contact length between ER and mitochondria in each genotype from **c**. **e** Quantification of ER lengths per field from TEM of wildtype and UBXD8 KO cells from **c**. **f** Quantification of number of mitochondria per field from TEM of wildtype and UBXD8 KO cells from **c**. Measurements in **d**–**f** are from $n$ = 3 biological replicates with WT = 50 cells from 65 fields and UBXD8 KO = 53 cells from 71 fields. OMM: Outer mitochondrial membrane. Data are means ± SEM. *, **, ***$P$ < 0.05, 0.01, 0.0001 respectively, one-way ANOVA with Tukey's multiple comparison test (**b**, **d**); two-tailed unpaired $t$ test with Welch's correction (**e**, **f**). Scale bar, 100 nm (**c**). Source data are provided as a Source data file.

---

proteome in UBXD8 KO cells suggests that perturbation in the abundance of numerous enzymes linked to lipid biosynthesis may underlie the dysregulation in contacts.

**Deletion of UBXD8 results in an altered cellular lipidome**
To better understand how cellular lipid metabolism may be impacted by loss of UBXD8, we measured the lipidome of wildtype and UBXD8 KO cells. We identified and quantitatively measured the relative levels of phospholipids (PLs) using liquid-chromatography high-resolution tandem mass spectrometry (LC-MS/MS) in whole cell extracts. Our analyses examined the major classes of PLs found in membranes (and synthesized at contacts) including phosphatidylcholine (PC), phosphatidylethanolamine (PE), phosphatidylserine (PS), and phosphatidylinositol (PI). We also examined one-tailed lysophospholipids, which are metabolic by-products of phospholipids (e.g., LPC, LPE, LPS, and LPI). We determined the concentration of 151 PLs in the whole cell fraction in UBXD8 KO cells relative to their concentration in wildtype cells. Of 62 PL species, two-thirds of the two-tailed PC and PE and one-tailed LPC and LPE species were significantly abundant ([log$_2$ KO:WT] >1 and $P$ value <0.05) in UBXD8 KO cells compared to wild-type. (Fig. 5a, Supplementary Fig. 5a, and Supplementary Dataset 2). The PC species that increased the most in the UBXD8 KO whole cell fractions contained one or two double bonds among the fatty acyl tails (i.e., PC (44:1), PC (44:2), PC (46:1), PC (46:2), PC (48:2), PC (50:2), and PC (52:2)) (Fig. 5a and Supplementary Dataset 2). MS/MS identification of the individual tails revealed that each of these lipids contained only saturated or monounsaturated fatty acids ranging in size from C16:0 to C32:1 and that lipids with one or fewer double bonds in each tail were

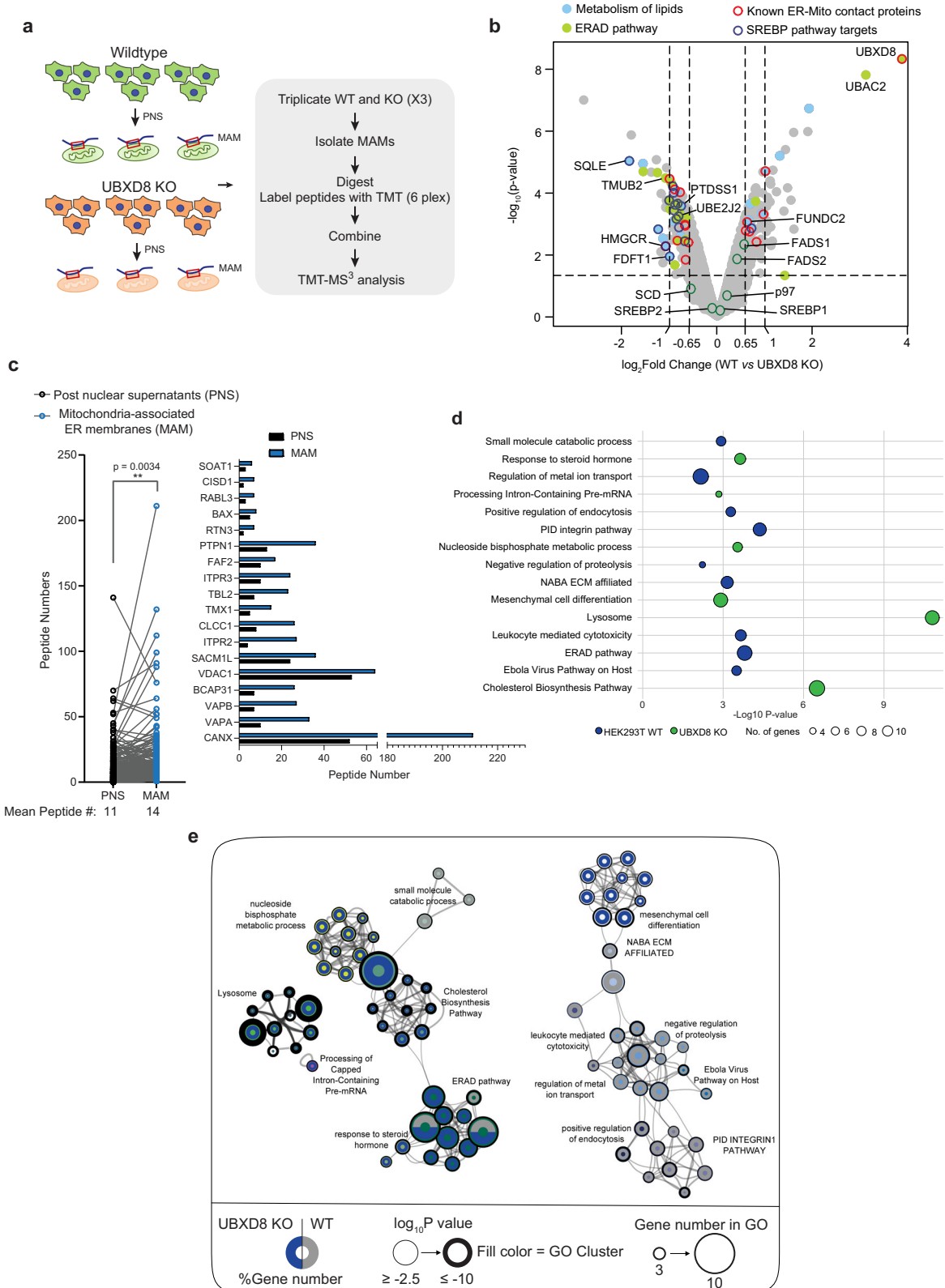

increased the most by the loss of UBXD8 (Fig. 5a, Supplementary Fig. 5, and Supplementary Dataset 2). Phospholipids are synthesized via a diacylglycerol (DG) intermediate and DGs are metabolized to generate PLs and TGs. We therefore extended our lipidomics studies to measure DG and two additional neutral lipids: triacylglycerol (TG) and cholesteryl esters (CE) that are major components of lipid droplets. The relative concentration of most DGs were altered by less than twofold (Supplementary Fig. 5b). Of the ten DG species increased by ≥2-fold in

UBXD8 KO cells, nine contained only saturated or monounsaturated tails ranging in size from C14:0 to C32:0. Similar to PC and PE, most TG species were ≥2-fold more abundant in UBXD8 KO cells relative to control cells (Supplementary Fig. 5c). The TG species whose abundance was enhanced the most by the loss of UBXD8 were also enriched in saturated and monounsaturated fatty acyl tails from C14:0 to C32:1. In contrast, most CE species was unaltered or slightly depleted by the loss of UBXD8 (Supplementary Fig. 5d). In summary, loss of

**Fig. 4 | Quantitative proteomics identifies a role for UBXD8 in regulating lipid metabolism at ERMCS. a** Schematic of tandem mass tag (TMT) proteomic workflow from wildtype and UBXD8 KO HEK293T cells. PNS post-nuclear supernatant, MAM mitochondria-associated membrane. **b** Volcano plot of the ($-\log_{10}$-transformed $P$ value vs the $\log_2$-transformed ratio of wildtype/UBXD8 KO) proteins identified from MAM fractions of HEK293T cells. $n = 3$ biologically independent samples for each genotype. $P$ values were determined by empirical Bayesian statistical methods (two-tailed $t$ test adjusted for multiple comparisons using Benjamini–Hochberg's correction method) using the *LIMMA* R package; for parameters, individual $P$ values and $q$ values, see Supplementary Dataset 1. **c** Left panel: Scatter plot of one-to-one comparison of the peptide numbers identified from TMT proteomics in (**b** and Supplementary Fig. 4b) for known MAM proteins between PNS (black circle) vs MAM (blue circle) fractions (\*\*$P < 0.01$, two-tailed paired $t$ test). Right: Bar graph of peptide numbers identified for well-established ERMCS proteins

identified in MAM fractions (blue bar) as compared to PNS fractions (black bar). **d** Bubble plot representing significantly enriched GO clusters identified from TMT proteomics of MAM fractions in wildtype (blue) or UBXD8 KO (green) cells (**a**, **b**, **e**). Size of the circle indicates the number of genes identified in each cluster. **e** Network of differentially enriched terms shown as clustered functional ontology categories. Each node represents a functional ontology term enriched in the TMT data (**a**, **b**) as scored by Metascape[86]. Networks were generated using Cytoscape v3.8.2. Size of node represents number of genes identified in each term by gene ontology (GO). Gray and blue donuts represent percent of genes identified in each GO term in wildtype or UBXD8 KO respectively. Node outline thickness represents $-\log_{10}$-transformed $P$ value of each term. The inner circle color of each node indicates the corresponding functional GO cluster. Default settings on Metascape were used to perform accumulative hypergeometric statistical test to calculate the $P$ values (**d**, **e**).

---

UBXD8 shifts the lipidome to have a greater abundance of PC, PE, DG, and TG with saturated and monounsaturated fatty acyl tails demonstrating that UBXD8 is necessary for regulating lipid concentrations. The synthesis of these phospholipids occurs at contacts and their altered abundance in UBXD8 KO cells may contribute to defects in contacts[36].

### Defective SREBP1 activation underlies ERMCS defects in p97-UBXD8 loss-of-function cells

The significantly altered lipidome and related alterations in lipid biosynthetic enzymes prompted us to ask if these changes stem from the regulation of the SREBP1/2 pathway. p97-UBXD8 mediates the activation of ER-localized SREBPs by mediating the membrane extraction and degradation of the SREBP negative regulator INSIG1 when it is ubiquitylated by GP78, in a sterol or fatty acid-dependent manner (see below)[37,38]. A similar pathway for lipid sensing in *S. cerevisiae*, requires Cdc48p and Ubx2p (orthologs of p97 and UBXD8, respectively)[39], and yeast lacking Ubx2p have more saturated cellular membranes due to loss of transcriptional activation of $\Delta^9$ desaturase *ole1*[39]. Indeed, the lipid desaturases FADS1 and FADS2 were modestly depleted in UBXD8 KO MAM fractions (Fig. 4b). We found that loss of p97 or UBXD8 resulted in a significant loss of SREBP1 activation and accumulation of the inactive ER-tethered form (Fig. 6a–d). We were unable to detect endogenous INSIG1 in cells due to its short half-life and low abundance[40] therefore, we transiently expressed HA-tagged INSIG1 and observed robust stabilization in p97-UBXD8 loss of function cells (Fig. 6a–d). These results were also reproduced in the GP78 KO cell line which is the E3 ligase that ubiquitylates INSIG1 (Fig. 6e). Lipid desaturases are transcriptional targets of SREBP1 and were decreased in abundance at the transcript level in UBXD8 KO cells (Supplementary Fig. 6a). This parallels the decreased protein abundance of the best characterized desaturases, SCD1 ($\Delta^9$) and FADS1 ($\Delta^5$) in UBXD8 KO or p97 depleted cells (Fig. 6a–d). In contrast, UBXD8 loss did not significantly impact SREBP2 activation or its downstream targets although this may be cell type or context specific (Fig. 6a, b and Supplementary Fig. 6b, c). Similar results were observed by fractionating cells and probing for nuclear SREBP1 (Supplementary Fig. 6d). Previous studies have reported that ERMCS have lipid raft-like properties and are enriched in cholesterol and sphingolipids[41], thus the localization of cholesterol-sensing proteins to these sub-domains may be advantageous for rapid pathway activation. We find that SREBP1 and SCD1 are enriched at ERMCS isolated by biochemical fractionation (Fig. 6f). Notably, an increase in ER-tethered SREBP1 and a decrease in SCD1 were apparent in the MAM fractions of UBXD8 KO cells relative to the rest of the ER suggesting that SREBP activation by p97-UBXD8 may take place preferentially at ERMCS (Fig. 6g, h). Furthermore, overexpression of mature SREBP1a and c and to a lesser extent SREBP2, is sufficient to rescue contacts and SCD1 protein levels in p97-UBXD8 depleted cells (Supplementary Fig. 6e–g). Hence, defective SREBP1 activation underlies increased contacts in p97-UBXD8-depleted cells.

The decrease in the abundance of lipid desaturases and our finding that UBXD8 KO cells have increased phospholipids with saturated or monounsaturated tails, prompted us to determine whether altering membrane lipid saturation perturbed contacts between the ER and mitochondria. FADS1 isoforms are localized to both ER and mitochondria[42], therefore we focused on ER-localized SCD1[43] as it was enriched at contacts (Fig. 6f). We treated wild type HEK-293T cells with the SCD1 inhibitor MF438[44] and found that contacts between the ER and mitochondria increased in a manner that could be rescued by supplementing cells with monounsaturated oleic acid, the product of SCD1, (Supplementary Fig. 6h). We next evaluated whether re-expressing SCD1 in p97 or UBXD8-depleted cells rescued the increased contact site phenotype. Overexpression of wildtype SCD1 rescued ER–Mitochondria contacts to wildtype levels in p97-UBXD8 depleted cells (Fig. 6i and Supplementary Fig. 6i). In contrast, a catalytically inactive version of SCD1 was unable to rescue the phenotype (Fig. 6i and Supplementary Fig. 6i). Notably, overexpression of a SCD1 catalytic mutant in wild type cells resulted in an increase in contacts suggesting that the resulting ordered lipid bilayers impacts contacts (Fig. 6i). To further extend these findings, we asked if simply supplementing p97-UBXD8 depleted cells with unsaturated oleic acid (18:1), a precursor for the generation of polyunsaturated fatty acids in cells was sufficient to rescue contacts. Indeed, oleic acid but not saturated palmitic acid (16:0) rescued the contact defect in p97-UBXD8 depleted cells. Strikingly, palmitic acid alone increased contacts in wildtype cells (Fig. 6j). Oleic acid and palmitic acid can differentially impact lipid droplet formation and mitochondrial respiration and fragmentation[45]. Thus, the impact of these lipids on ERMCS may be due to indirectly impacting lipid droplets or mitochondria morphology. UBXD8 has previously been reported to partition into lipid droplets as they bud from the ER where it serves to recruit p97 to inhibit the major lipase ATGL by dissociating it from its co-activator CGI-58[26,46]. Furthermore, we and others find a fraction of UBXD8 on mitochondria[20]. Lipid droplets (labeled with BODIPY 493/503) were smaller in UBXD8 KO cells in agreement with a role for UBXD8 in preventing lipolysis, but their numbers were unchanged when compared to wildtype cells (Supplementary Fig. 7a–c). Furthermore, oleic and palmitic acid treatment resulted in a comparable increase in lipid droplet number and size in both wildtype and UBXD8 KO cells suggesting that the ability of these lipids to differentially impact ERMCS in wildtype and UBXD8 KO cells is not likely due to changes in lipid droplets under these conditions (Supplementary Fig. 7a–c). We next imaged mitochondria labeled with TOMM20 in wildtype and UBXD8 KO cells treated with oleic or palmitic acid and quantified the mitochondrial area and network morphology using mitochondrial network analysis (MiNA)[47]. Antimycin and oligomycin co-treatment was used as a positive control in wildtype cells to collapse the inner mitochondrial membrane potential and cause fragmentation (Supplementary Fig. 7d–g). Unlike antimycin-oligomycin treatment which caused significant mitochondrial fragmentation, oleic acid and palmitic acid treatment induced very minor

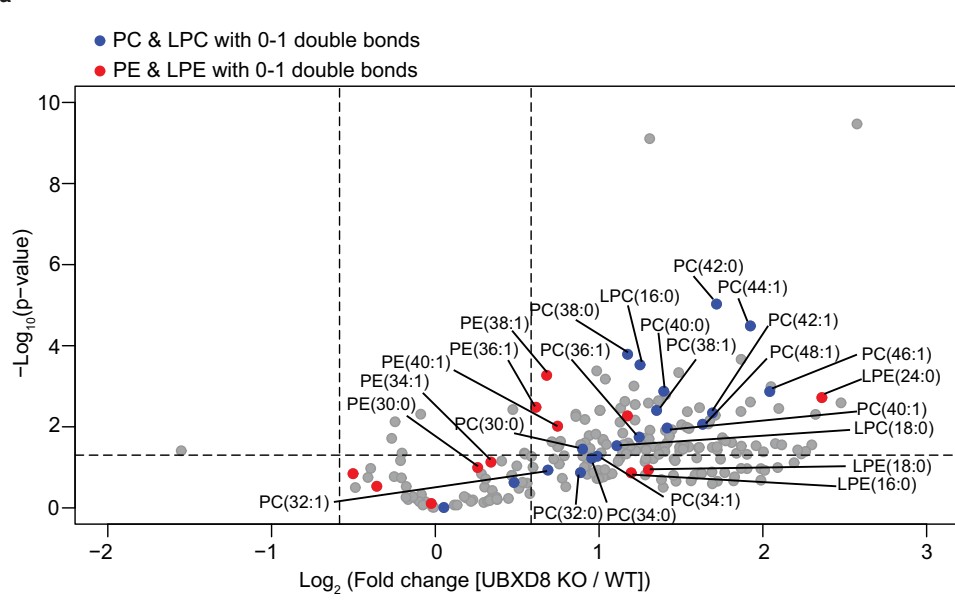

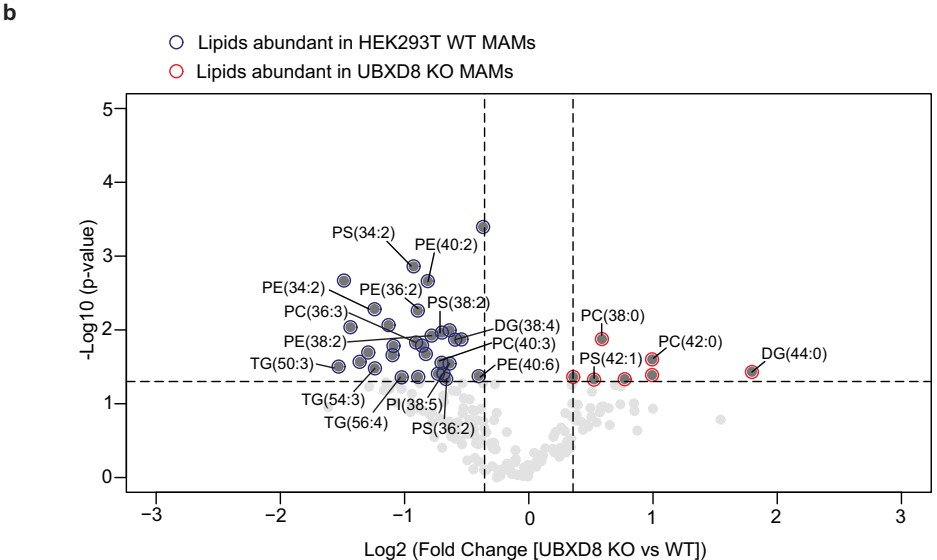

**Fig. 5 | Quantitative lipidomics of MAMs identifies a role for UBXD8 in regulating fatty acid composition at ERMCS. a**, **b** Volcano plot of the (−log$_{10}$ transformed *P* value vs the log$_2$-transformed ratio of UBXD8 KO: wildtype) total phospho- and neutral lipid species identified using lipidomics of whole cell extracts (**a**) or MAM fractions (**b**) of HEK293T cells. PC, LPC species (blue filled circles) and PE, LPE species (red filled circles) with saturated or monounsaturated fatty acid tails are labeled in **a**. Lipid species with saturated or monounsaturated fatty acid tails (red outline) or polyunsaturated fatty acid tails (blue outline) are labeled in the lipidomics of MAM fractions in **b**. Lipids were measured by LC-MS/MS following normalization by total protein amount. (*n* ≥ 3 biologically independent experiments were performed, each with duplicate samples). Statistical analysis was performed on the log transformed relative fold change values (UBXD8 KO relative to WT) using independent two-tailed *t* tests and Benjamini–Hochberg correction in R stats package (*P* values are listed in Supplemental Dataset 2). PC phosphatidyl choline, LPC lysophosphatidyl choline, LPE lysophosphatidyl ethanolamine, PE phosphatidyl ethanolamine, PI phosphatidylinositol, PS phosphatidylserine, DG diacylglycerol, TG triacylglycerol.

effects under the conditions tested and the changes were comparable between wildtype and UBXD8 KO cells (Supplementary Fig. 7d–g).

In summary, we conclude that p97-UBXD8 regulate SREBP1 activation and SCD1 protein levels to maintain the normal repertoire of ERMCS.

**Cellular membrane fluidity is dependent on p97-UBXD8**
Collectively, these results suggested that contacts are exquisitely sensitive to perturbations in lipid profiles within cellular membranes and that loss of p97-UBXD8 alters membrane lipid composition and saturation. We therefore asked how membrane lipids were altered

specifically in the MAM fraction in UBXD8 KO cells. We isolated MAM fractions in quadruplicate from wildtype and UBXD8 KO cells and determined the relative changes in lipid levels as detailed above. Of the 195 lipids identified, 37 lipids were significantly changed in the MAM fraction of UBXD8 KO cells (Fig. 5b and Supplementary Dataset 2). Unlike our findings from whole cell extracts, only seven lipids were increased in UBXD8 KO cells, whereas 30 were depleted. There were no obvious trends in specific classes of lipids changing, however, six of the seven lipids that were increased in UBXD8 KO cells, were saturated or monounsaturated (Fig. 5b and Supplementary Dataset 2). Furthermore, 22 of the 30 lipids that were depleted in UBXD8 KO MAM

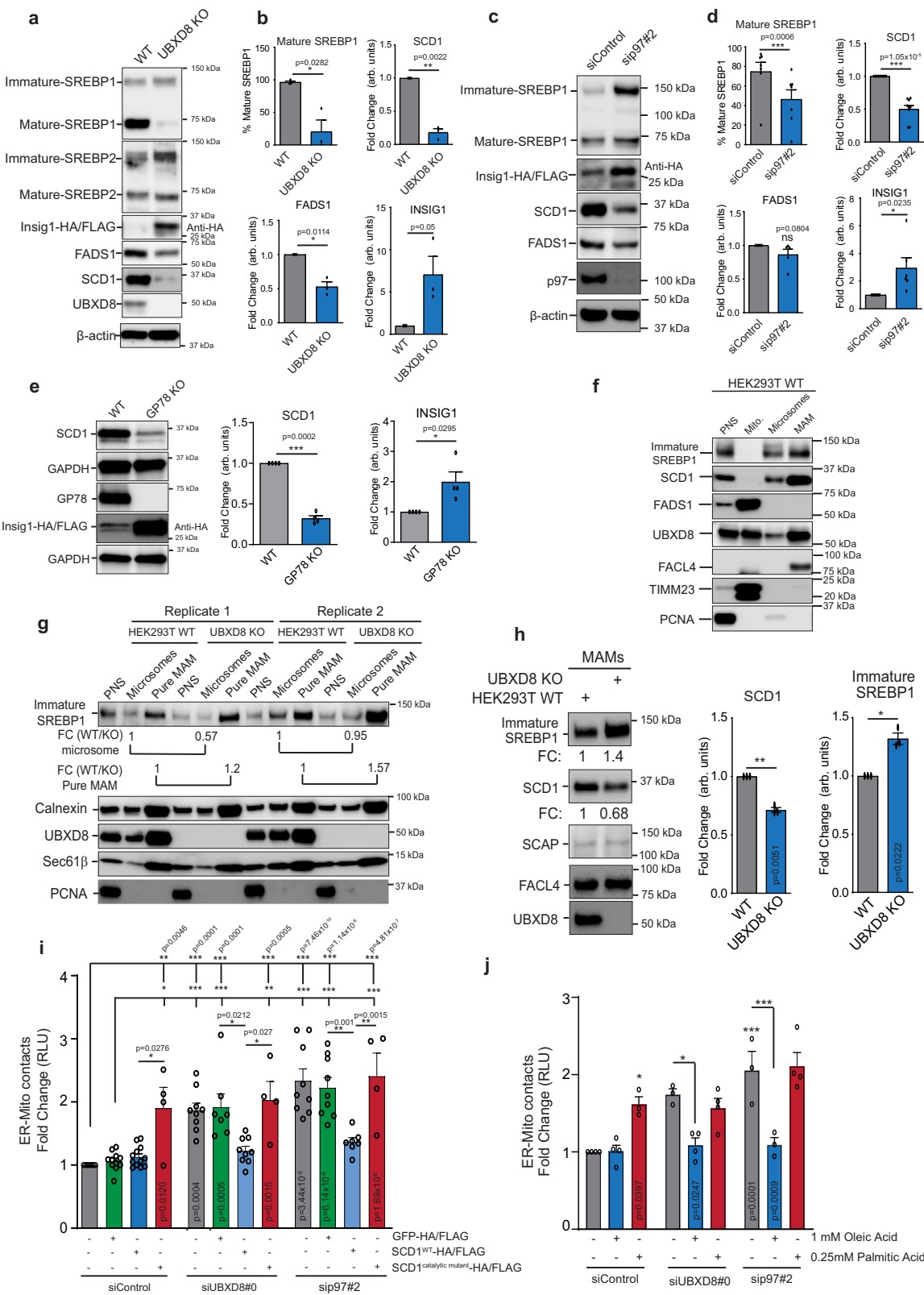

fractions were polyunsaturated (Fig. 5b and Supplementary Dataset 2). These findings suggest that ERMCS are more ordered in UBXD8 KO cells.

While we find that SCD1 is enriched in MAM fractions, unsaturated lipids produced by SCD1, and other desaturases are trafficked to membranes throughout the cell. We next evaluated whether loss of p97-UBXD8 impacts lipid saturation globally within the cell. We used lipid-binding pyrene probes that insert into both disordered

(unsaturated) and ordered (saturated) lipid bilayers throughout the cell and undergo monomer to excimer formation in a manner dependent on local membrane order. We measured the ratio of monomer to excimer fluorescence as an indicator of membrane order in cells treated with the p97 inhibitor CB-5083 or in UBXD8 KO cells. Loss of UBXD8 or inhibition of p97 resulted in more ordered cellular membranes (monomer: excimer ratio <1) compared to controls in HEK293T and HeLa cells (Fig. 7a, b). Strikingly this phenotype can be

**Fig. 6 | Loss of SREBP1 activation and SCD1 expression upon p97-UBXD8 depletion is responsible to ERMCS defects. a–e** Immunoblot and the corresponding band intensity quantifications of indicated proteins in the SREBP pathway in wildtype and UBXD8 KO HEK293T cells (**a**, **b**), p97-siRNA depleted cells (**c**, **d**) or HEK293 WT and GP78 KO cells (**e**). All samples were transfected with INSIG1-HA/FLAG due to lack of reliable antibodies to the endogenous protein. (For **b**: $n = 3$; for **d**: $n = 7$ (mature SREBP1), $n = 9$ (SCD1), and $n = 5$ (FADS1 and INSIG1); for **e**: $n = 4$ biologically independent samples). **f** Immunoblot of indicated SREBP pathway proteins from subcellular fractionation of HEK293T cells. PNS post-nuclear supernatant, Mito mitochondria, MAM mitochondria-associated membrane. ($n = 3$ biologically independent samples). **g** Immunoblot of indicated proteins from subcellular fractionation of HEK293T cells. PNS post-nuclear supernatant, MAM mitochondria-associated membrane. Corresponding fold changes in band intensities (FC: WT vs UBXD8 KO) of immature SREBP1 normalized to calnexin is shown below immature SREBP1 blot ($n = 2$ biologically independent samples).

**h** Immunoblot of indicated SREBP pathway proteins from subcellular fractionation of wildtype and UBXD8 KO HEK293T cells, MAM mitochondria-associated membrane. ($n = 3$ biologically independent samples). Corresponding fold changes in band intensities (FC: UBXD8 KO vs WT) of SREBP1 and SCD1 normalized to FACL4 is shown. **i** Split luciferase assay in HEK293T cells transfected with the indicated siRNAs and wildtype or catalytically dead SCD1. GFP-HA/FLAG was transfected as a negative control. RLU relative luminescence unit. ($n = 11, 11, 12, 4, 9, 7, 9, 4, 9, 9, 7$, and 4 biologically independent samples from left to right, respectively). **j** Split luciferase assay in HEK293T cells transfected with the indicated siRNAs and treated with either monounsaturated oleic acid or saturated palmitic acid. RLU relative luminescence unit. ($n = 4, 4, 3, 3, 4, 4, 3, 3$, and 4 biologically independent samples from left to right, respectively). Data are means ± SEM. *, **, ***$P < 0.05, 0.01$, 0.0001, respectively. One-tailed paired $t$ test (**b**, **d**, **e**), two-tailed paired $t$ test (**h**) or one-way ANOVA with Tukey's multiple comparison test (**i**, **j**). Source data are provided as a Source data file.

reversed by incubating cells with oleic but not palmitic acid (Fig. 7a, b). These findings prompted us to investigate whether the global increase in membrane order impacts other ER-organelle contacts. We used a reporter construct previously shown to localize to ER-plasma membrane (PM) junctions known as membrane-attached peripheral ER (MAPPER) which consists of the signal peptide and transmembrane domain of the ER Ca²⁺ sensor STIM1 fused to GFP followed by a polybasic motif that binds to phosphoinositides in the PM[48]. Cells expressing this marker were depleted of p97 and UBXD8 with siRNAs or treated with CB-5083 and the GFP signal was quantified. Compared to control samples, loss of p97-UBXD8 or inactivation of p97 enzymatic activity resulted in a significant increase in GFP-MAPPER puncta indicating an increase in ER-PM contacts (Fig. 7c, d). This finding was further supported by analysis of TEM data from wildtype and UBXD8 KO cells which also demonstrated increased ER–PM contacts (Fig. 7e, f). Thus, the degradation of INSIG1 and subsequent activation of SREBP1 and SCD1 impacts lipid composition and saturation throughout the cell but significantly impact contacts due to their reliance on membranes for association.

### Increased ERMCS and inactivation of SREBP1 and SCD1 observed in mouse models of p97 disease

Mutations in p97 cause several primarily neurodegenerative protein aggregation disorders. These include inclusion body myopathy with Paget's disease of the bone and frontotemporal dementia (IBMPFD, also known as multi-system proteinopathy 1, MSP-1)[49], amyotrophic lateral sclerosis (ALS)[50,51], Charcot Marie Type IIB[52], among others. We investigated whether p97 disease-associated mutations altered contacts between the ER and mitochondria and perturbed the SREBP1-SCD1 pathway. We measured contacts in mouse embryonic fibroblasts heterozygous or homozygous for p97 R155H (a prevalent mutation observed in patients) and observed that cells with p97 R155H homozygous mutation had a significant increase in contacts that could be rescued with oleic acid but not palmitic acid (Fig. 8a).

Next, we evaluated the SREBP-SCD1 pathway in the brains of the two distinct p97 mouse models[53]. A conditional knockout of p97 (p97 cKO) in the cortex and hippocampus has recently been shown to develop cortical atrophy, neuronal loss and TDP43 inclusions reminiscent of frontotemporal dementia[53]. We stained for SREBP1 and SCD1 using siRNA-validated antibodies in the CA1 regions of one-month-old p97 cKO mice before neurodegeneration phenotypes are observed (Fig. 8b and Supplementary Fig. 8). Strikingly, p97 cKO mice had a significant decrease of SREBP1 and SCD1 immunoreactivity in the CA1 region compared to age-matched controls (Fig. 8b, c). To assess whether these defects were also present in a pathogenic context, we immunoblotted for SREBP1 and SCD1 in brain lysates from 6- and 13-month-old p97[R155C/WT] mice[53,54]. SREBP1 processing was significantly diminished at 13 months in p97[R155C/WT] mice relative to controls (Fig. 8d, e). Similarly, loss of SCD1 protein levels was apparent at 6 months and

continued to decline at 13 months in p97[R155C/WT] mice (Fig. 8d, e). Collectively, our findings suggest that p97 mutations that cause disease may also have underlying lipid metabolism deficits that could contribute to disease pathology.

## Discussion

Here we have identified an unanticipated role for p97 and its ER-tethered adapter UBXD8 in regulating ER-mitochondria contacts by perturbing membrane composition and fluidity in multiple cell types. Our studies suggest that UBXD8 is unique among p97 adapters in that it is enriched at ERMCS and recruits p97 to contacts (Fig. 1). Loss of p97-UBXD8 results in an increase in contacts as measured using multiple ERMCS reporters, including split-luciferase and split-GFP assays, and TEM studies (Figs. 2 and 3). The ability to maintain the normal number of contacts is dependent on the formation of a p97-UBXD8 complex and the presence of a ubiquitylated substrate (Fig. 2 and Supplementary Fig 1c). The INSIG-SCAP-SREBP pathway is a known target of p97-UBXD8. UBXD8 recruits p97 to mediate the extraction and degradation of INSIGs when they are ubiquitylated by the E3 ligase GP78. Loss of INSIGs which are the negative regulator of SREBPs, allows SCAP to escort SREBPs to the Golgi where SREBPs are cleaved by proteases to release the soluble transcription factor that can turn on numerous genes involved in cholesterol and lipid biosynthesis (Fig. 8f). Notably, lipid desaturases are key targets of this pathway and of relevance to this study.

Thus, through the SREBP pathway p97-UBXD8 can regulate the lipidome in diverse ways. Indeed, this is borne out in our proteomic and lipidomic analysis that found significantly altered levels of lipid biosynthetic proteins and different classes of phospholipids respectively (Figs. 4 and 5). Our proteomic studies identified multiple desaturases in the MAM fractions including SCD1 ($\log_2$FC WT:UBXD8 KO = −0.54), FADS1 ($\log_2$FC WT:UBXD8 KO = 0.60) and FADS2 ($\log_2$FC WT:UBXD8 KO = 0.42) (Fig. 4b and Supplementary Dataset 1). We note that while our biochemical studies assessing SREBP1-SCD1 demonstrated significant changes in these proteins (Fig. 6), their levels did not significantly change in the TMT study (Fig. 4b). This is likely due to ratio compression of the quantified peptides in the proteomic studies (see Methods for further discussion).

In agreement with the role of p97-UBXD8 in activating the SREBP pathway, our studies find that loss of p97, UBXD8 or GP78 leads to stabilization of INSIG1, decreased cleavage of SREBP1 and diminished levels of the lipid desaturases SCD1 and FADS1 (Fig. 6). Importantly, under these conditions we also observe an increase in ERMCS that can be rescued by complementing these cells with the cleaved (transcription factor form) of SREBPs and wildtype (but not catalytically inactive) SCD1 (Fig. 6). These studies suggest that loss of SCD1 and the resulting loss of lipid desaturation cause the increase in ERMCS upon loss of p97-UBXD8. Notably, both full-length ER-localized SREBP1 and SCD1 are significantly enriched in MAM fractions and loss of

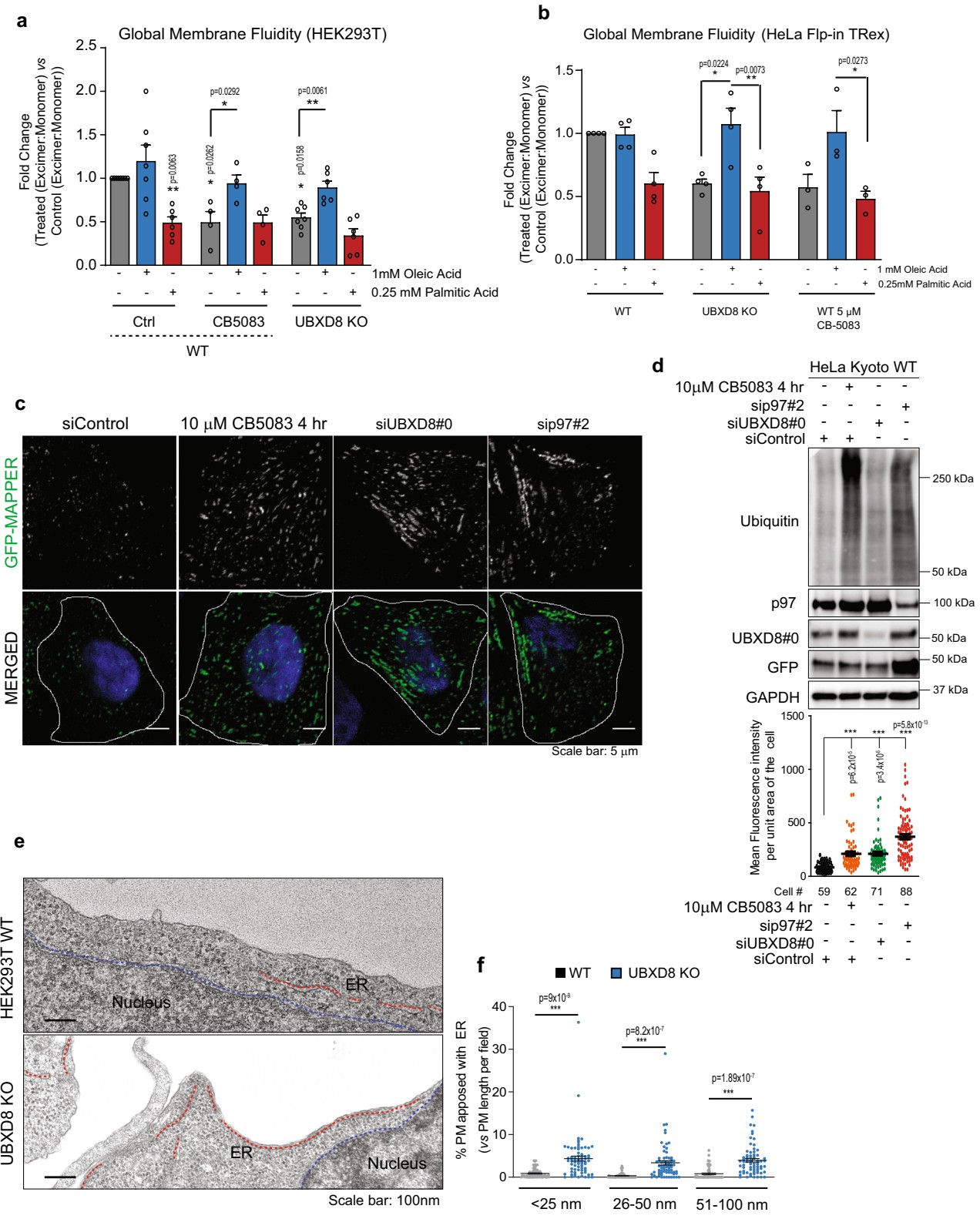

UBXD8 results in the increase in full length SREBP1 (due to decreased processing) and decreased SCD1 in these fractions relative to the rest of the ER (Fig. 6g). In further support of these findings, lipidomics of MAM fractions cells found an enrichment of saturated and mono-unsaturated lipids and a depletion of polyunsaturated lipids in UBXD8 KO cells relative to wildtype cells. We propose that p97-UBXD8 regulates the activation SREBP1 at ERMCS relative to the rest of the ER (Fig. 6g). Intriguingly, our studies using lipid disorder probes indicate

that increased membrane saturation is not exclusive to ERMCS but is found throughout cellular membranes (Fig. 7). Consequently, we find that ER-PM contacts are also increased upon p97-UBXD8 loss of function (Fig. 7). Whether other organelle contacts are similarly impacted due to increased membrane order is a topic of future study. We propose that altered lipid bilayers that arise upon loss of UBXD8 impact contacts in multiple ways: (1) preventing their dynamic association and disassociation due to loss of fluidity, and (2) negatively

**Fig. 7 | Depletion of p97-UBXD8 alters global membrane fluidity and ER–PM contacts. a, b** Global membrane fluidity was measured using a pyrene-based lipid probe in wildtype and UBXD8 KO HEK293T cells (**a**) or HeLa-FlpIN-TRex cells (**b**). Wildtype cells were also treated with 5 μM of the p97 inhibitor CB-5083 for 4 h. Cells were supplemented with indicated concentrations of oleic acid and palmitic acid for 4 h. The fold change (Treated$_{excimer:monomer}$ vs Control$_{excimer:monomer}$) of the ratio of excimer (Em. Max. 460 nm) to monomer (Em max. 400 nm) fluorescence is indicated. Fold changes <1 indicate more ordered lipid bilayers relative to wildtype untreated control. (For **a**: $n = 8, 7, 6, 4, 4, 4, 7, 6,$ and 6 biologically independent samples from left to right, respectively; for **b**: $n = 4, 4, 4, 4, 4, 3, 3,$ and 3 biologically independent samples from left to right, respectively). **c** Representative fluorescence microscopy images showing the ER–PM contact sites reporter, GFP-MAPPER in HeLa Kyoto cells transfected with the indicated siRNAs for 48 h or treated with 10 μM CB5083 for 4 h. **d** Top: Immunoblot of indicated proteins; Bottom: Quantification of mean fluorescence intensity of GFP-MAPPER per unit area of cell for **c**. **e** Representative transmission EM micrographs of wildtype and UBXD8 KO HEK293T cells illustrating contacts between ER (red dotted line) and plasma membrane. Nucleus boundary is marked by blue dotted line. **f** Quantification of contact length between ER and plasma membrane in each genotype from **e** (measurements are from $n = 3$ biological replicates with WT = 50 cells from 65 fields and UBXD8 KO = 53 cells from 71 fields). Data are means ± SEM. *, **, ***$P < 0.05, 0.01, 0.0001$, respectively. Significance was analyzed by one-way ANOVA with Tukey's multiple comparison test (**a, b, d, f**) or one-tailed Student's $t$ test for columns in **a**. Source data are provided as a Source data file.

impacting the lateral movement of tethering proteins within membranes. A recent study using high speed molecular tracking of the ERMCS tether VAPB found that VAPB diffused rapidly in and out of contact sites, but that VAPB molecules within contacts displayed significantly decreased diffusion within contacts relative to the surrounding ER[55]. What factors dictate reduced diffusion are not presently known, but our studies suggest that lipid order may have a significant impact on the stability of contacts and/or the diffusion rates of tethers.

Given the significant changes in lipid profiles in UBXD8 KO cells, it is likely that SCD1 regulation is not the only mechanism at play. It remains to be determined whether UBXD8 also facilitates the degradation of contact tethers. Indeed, p97 has been shown to regulate contacts through the degradation of MFN2 and by interacting with VPS13D[56] although it is not clear if this is dependent on UBXD8. We find no role for UFD1-NPL4 in regulating contacts, however this is not surprising as UFD1-NPL4 are not involved in the degradation of INSIG1. Aberrant contact sites are emerging as a common feature in the pathophysiology of a wide spectrum of human diseases ranging from diabetes to neurodegeneration[57–59]. We find that p97 mutations that cause proteinopathies also exhibit increased contacts and display significantly decreased levels of SREBP1 and SCD1. A recent report found that motor neurons from ALS patients with p97 mutations exhibited more contacts between the ER and mitochondria relative to controls[60]. Furthermore, several studies suggest that a lipid-controlled diet or modulation of lipid biosynthesis may have therapeutic implications in mouse models of p97 proteinopathies[61,62]. Thus, altered organelles contacts and downstream lipid synthesis warrants further investigation in p97 associated diseases.

## Methods

### Ethics statement

Animal procedures were performed in accordance with protocols approved by the Institutional Animal Care and Use Committee (Animal Welfare Assurance # D16-00245; Protocol No. 22-0298) at Washington University School of Medicine.

### Cell culture, transfections, immunoprecipitations, and treatments

HEK293T (ATCC# CRL-3216™), HEK293 WT (ATCC# CRL-1573™) and GP78 KO (gift from James Olzmann, University of California Berkeley), HeLa Kyoto (CLS Cell Lines Service GmbH; catalog number 300670; gift from Ron Kopito, Stanford University), Mouse embryonic fibroblasts (MEFs), COS7 cells (ATCC# CRL-1651™), and HeLa-Flp-IN-TREX (HFTs (ThermoFisher Cat# R71407) with introduced Flp-In site (Flp-In™ T-REx™ Core Kit, Cat# K650001; Thermofisher Scientific is a gift from Brian Raught, University of Toronto) cells were cultured in Dulbecco's modified Eagle's medium, supplemented with 10% fetal bovine serum (FBS) and 100 units/ml penicillin and streptomycin. Cells were maintained in a humidified, 5% $CO_2$ atmosphere at 37 °C.

For siRNA transfections, HEK293T cells were trypsinized and reverse transfected with siRNAs. HeLa Kyoto, MEF, and HFT cells were trypsinized and seeded into a 12- or 6-well dish 24 h prior to siRNA transfections. In both reverse and forward transfections, the cells were transfected with 20 nM siRNAs using RNAiMax (Invitrogen) according to the manufacturer's protocol. Cells were harvested 48–72 h post transfection. For cDNA transfections, HEK293T cells in 6-well plates were transfected with 0.75 μg each of pcDNA3-Mit-NRluc91 and pcDNA3-Crluc92-ER, 0.75 μg of UBXD8-C-HA/FLAG constructs, 1 μg of N-Myc-p97 constructs, 1 μg of p97-C-Myc, 1 μg of SCD1-C-HA/FLAG constructs, 0.2–0.5 μg of GFP-C-HA/FLAG, 0.75 μg of pCIG construct, 0.75 μg of REEP1, 0.75 μg each of pCI-Neo-PTPIP51-HA and pCI-Neo-Myc-VAPB or 0.75 μg each of 2X-FLAG-SREBP1a, 2X-FLAG-SREBP1c, 2X-FLAG-SREBP2 using Polyehtylenimine (PEI) at 1:4 DNA:PEI ratio and typically harvested 36–48 h post transfection. HeLa Kyoto, MEF, Cos7, and HFT cells were transfected with 0.4 μg of GFP-MAPPER (Addgene plasmid # 117721), 0.25 μg each of mitochondrial localizing OMM-GFP$_{1-10}$ and ER localizing short GFP$_{\beta 11}$, 1 μg of pHAGE-Sec61β-C-eGFP, 0.5 μg of Mito-BFP, 1 μg of pHAGE-mCherry-UBXD8, or 0.5 μg of Sec61β-mCherry using Lipofectamine 2000 (Invitrogen) and the cells were harvested 36–48 h post transfection. Cells were lysed in mammalian cell lysis buffer (50 mM Tris-Cl, pH 6.8, 150 mM NaCl, 0.5% Nonidet P-40, HALT Protease inhibitors (Pierce) and 1 mM DTT). Cells were incubated at 4 °C for 10 min and then centrifuged at 19,000 × g for 15 min at 4 °C. The supernatant was collected, and protein concentration was estimated using the DC protein assay kit (Biorad). Protein G agarose (Pierce, Thermo Fisher scientific) and the indicated antibodies were used for immunoprecipitation at 4 °C for 3–5 h. Beads were washed 3–5 times in 1 ml mammalian cell lysis buffer and resuspended in 2× SDS sample buffer. Cells were treated with 1 μM of Bortezomib, 5 μM CB-5083, 5 μg/ml of puromycin, 10 μM of Antimycin, 5 μM Oligomycin, 10 μM Brefeldin A, 1.5 mM Dithiothreitol, 2.5 μM Tunicamycin, 1.5 μM Thapsigargin, 1 μM SCD1 inhibitor (MF438), 1 mM Oleic acid, or 0.25 mM Palmitic acid for the indicated times (see figures for details). A full list of constructs used in this study can be found in Supplementary Table 1.

### Generation of CRISPR cell lines

The CRISPR-Cas9 gene editing system was used to generate UBXD8 knockout cell lines in HEK293T, and HFT cells. The guide sequence 5′ GTTAACCTGCAGGGGTCGTGA 3′ was cloned into the pX459 vector carrying the hSpCas9 and transiently transfected into HEK293T and HeLa-Flp-IN-TRex cells using Lipofectamine 3000 (Invitrogen) per the manufacturer's protocol. 36 hours post-transfection the cells were selected with 1 μg/ml puromycin for a further 24–36 h. The surviving cells were then serially diluted into 96-well plates for clonal selection and expression levels were monitored by immunoblotting.

### Antibodies, siRNAs and reagents

The p97 (10736-1-AP; WB: 1:2000), UBXD8 (16251-1-AP; WB: 1:2000), FACL4 (22401-1-AP; WB: 1:2000), UBXD2 (21052-1-AP; WB: 1:2000), HRD1 (13473-1-AP; WB: 1:2000), Sec61β (51020-2-AP; WB: 1:2000), Calnexin (10427-2-AP; WB: 1:2000), UBXN1 (16135-1-AP; WB: 1:3000), SREBP1 (14088-1-AP; WB: 1:2000), SREBP2 (28212-1-AP; WB: 1:2000), FADS1 (10627-1-AP; WB: 1:2000), anti-GFP (66002-1-AP; WB: 1:2000),

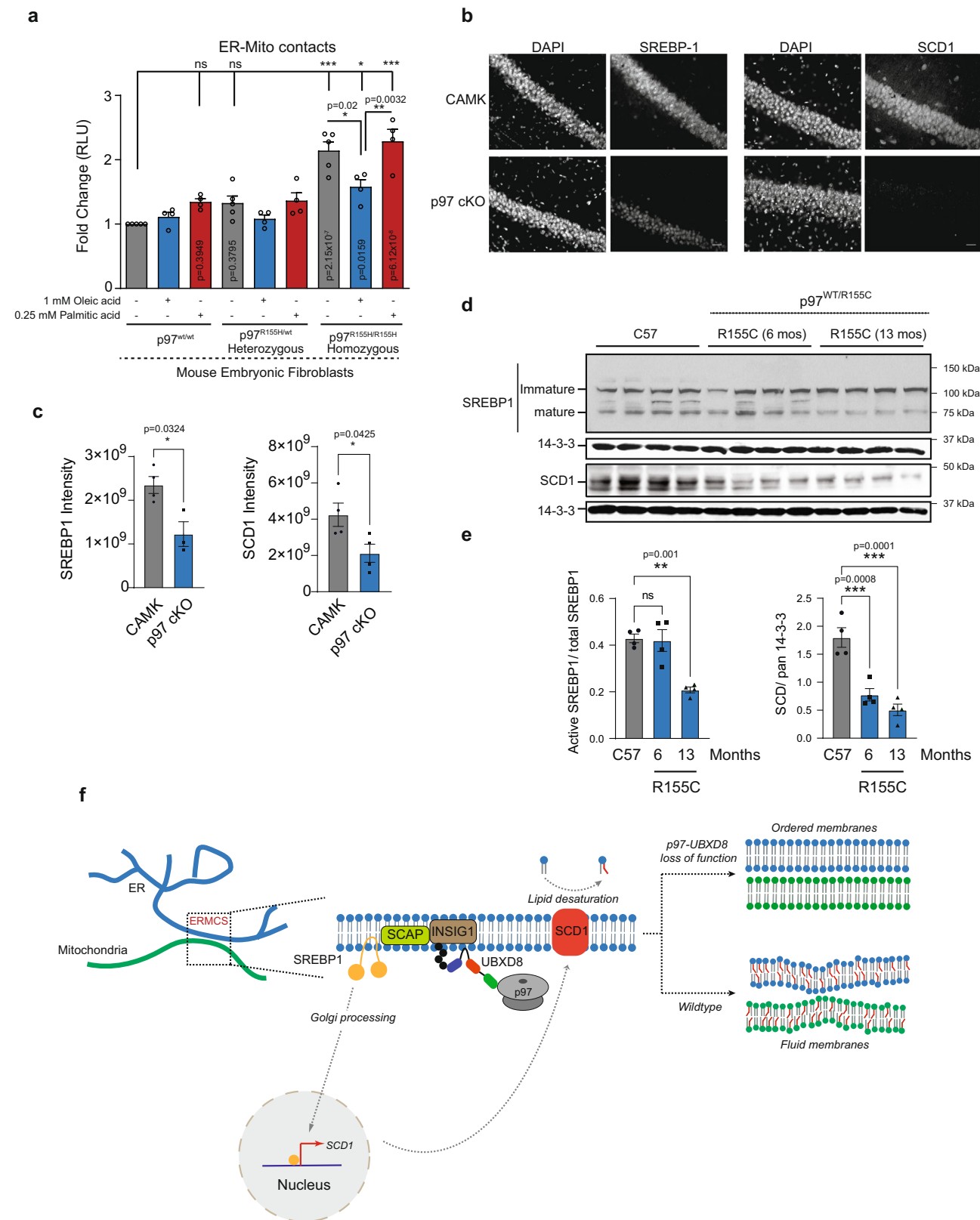

AMFR/GP78 (16675-1-AP; WB: 1:2000), GRP75 (14887-1-AP; WB: 1:2000), VAPB (14477-1-AP; WB: 1:2000), and SCD1 (23393-1-AP; WB: 1:2000) antibodies were from Proteintech Inc. The TIMM23 (H-8; sc514463; WB: 1:2000), TOMM20 (F-10; sc17764; WB: 1:2000; IF: 1:100), TOMM70 (A-8; sc390545; WB: 1:2000), pan-ubiquitin (P4D1; sc8017; WB: 1:2000), c-Myc (9E10; sc40; WB: 1:2000), β-Actin (AC-15; sc69879; WB: 1:2000), SIGMA1R (B-5; sc137075; WB: 1:2000), GAPDH (O411;

sc47724; WB: 1:2000), and PCNA (PC10; sc56; WB: 1:2000) antibodies were obtained from Santa Cruz Biotechnologies. LC3B (D11; 3868S; WB: 1:1000), and BiP (C50B12; 3177T; WB: 1:2000) were from Cell Signaling Technologies. p97 (A300-589A; WB: 1:2000) was from Bethyl Laboratories. The following antibodies Histone-H3 (ab1791; Abcam; WB: 1:2000), UBXD7 (PA5-61972; Invitrogen; WB: 1:2000), anti-HA (16B12; MMS-101P, Covance; WB: 1:2500), anti-FLAG (M2; F3165 Sigma

**Fig. 8 | p97 R155H disease mutation and conditional p97 knock out in mice recapitulates the p97-UBXD8 loss of function effect on ERMCS. a** Split luciferase assay to measure contacts in mouse embryonic fibroblasts with heterozygous or homozygous p97 R155H mutation. Cells were supplemented with indicated concentrations of oleic acid and palmitic acid for 4 h. (For **a**: $n = 5, 4, 4, 5, 4, 4, 5, 4$, and 4 biologically independent samples from left to right, respectively). **b** Representative SREBP1 and SCD1 staining from CA1 regions of 1 month-old control (CAMK2α) and p97 cKO mice (scale bar is 25 μm). **c** Quantification of fluorescence intensity of images in **b**. Individual points represent mean ROI intensity from each mouse, 3 or 4 animals per group. **d** Representative immunoblot for SREBP1 and SCD1 from cortical brain lysates of 12-month-old control (C57), or 6- and 12-month-old p97[R155C/WT] mice ($n = 4$ for each group, except for p97 cKO in SREBP1 panel ($n = 3$)). Pan 14-3-3 was used as housekeeping control. **e** Quantification of **d**. The ratio of mature SREBP1 to total SREBP1 is shown. SCD1 intensities are normalized to 14-3-3 levels in each lane. Individual points represent each mouse, 4 animals per group. Data are means ± SEM (\*, \*\*, \*\*\*$P < 0.05, 0.01, 0.0001$, respectively). Significance was analyzed by one-way ANOVA with Tukey's multiple comparison test (**a**) or Dunnett's multiple comparison (**e**) or two-tailed Student's $t$ test (**c**). **f** Model: Schematic of SREBP pathway activation. At ERMCS, UBXD8 recruits p97 to mediate the extraction and degradation of INSIG1 when it is ubiquitylated by the E3 ligase GP78. Loss of INSIG1, which is a negative regulator of SREBP1, allows translocation of cleaved SREBP1 transcription factor into the nucleus to activate expression of lipid desaturases, notably SCD1. Loss of p97-UBXD8 leads to decrease in SCD1 levels and loss of lipid desaturation resulting in membrane order and increased contacts. Source data are provided as a Source data file.

Aldrich; WB: 1:5000), were used for immunoblotting. HRP conjugated anti-rabbit (W401B; WB: 1:10,000) and anti-mouse (W402B; WB: 1:10,000) secondary antibodies were from Promega. Goat anti-Mouse IgG (H + L) Cross-Adsorbed Secondary Antibody, Alexa Fluor™ 568 (Catalog # A-11004; IF: 1:10,000), and Goat anti-Mouse IgG (H + L) Cross-Adsorbed Secondary Antibody, Alexa Fluor™ 488 (Catalog # A-11001; IF: 1:10,000) were purchased from Thermofisher Scientific. CB-5083 was a gift from Cleave Biosciences and Bortezomib was from Selleckchem. Palmitic acid (100905) is from MP Biomedicals and Oleic acid (270290050) is from Acros Organics. MF438 (569406) is from Millipore Sigma. BODIPY 493/503 (item ono. 25892) is from Cayman chemical company. All siRNAs were purchased from Ambion (Thermo Fisher Scientific): UBXD8-0 (s23260), UBXD8-9 (s23259), UBXD7-7 (s24997), NPL4-0 (s31210), NPL4-9 (s31209), and UFD1-7 (s14637). UBXD2-1 (D-014184-03), UBXD2-2 (D-014184-04), HRD1-3 (D-007090-03), and HRD1-4 (D-007090-04) were purchased from GE Dharmacon. siControl (SIC001) was from Millipore Sigma. p97 siRNAs (2-HSS111263 and 3-HSS111264), SCD1 siRNA (5′-GAUAUGCUGUGGUGCUUAA-3′)[63], and SREBP1 siRNA (5′-AUCUCUGAAGGAUCUGGUG-3′)[64] were from Invitrogen (Thermo Fisher Scientific). p97 rescue constructs were previously published[65] and were resistant to siRNA # 2. UBXD8-C-HA/FLAG construct was previously published[65]. The UBXD8 rescue constructs, including UBA\* ([17]LLQF[20] mutated to [17]AAAA[20]), ΔUAS (deleted amino acids between 122-277), and UBX\* ([407]FPR[409] mutated to [407]AAA[409]), were cloned using overlap PCR followed by Gibson assembly (NEB) cloning into pHAGE-C-HA/FLAG and were resistant to siRNA # 0. The SCD1-C-HA/FLAG WT and catalytic dead mutant (His[160] His[161] and His[301] His[302] mutated to Ala[160] Ala[161] and Ala[301] Ala[302]) constructs were cloned using overlap PCR followed by Gateway cloning (Thermo Fisher Scientific) into pHAGE-C-HA/FLAG.

## MAM fractionation

MAMs were isolated as previously described[15,66]. Briefly, HEK293T or HeLa-Flp-IN-T-Rex cells were seeded into four 150 mm TC dishes. Cells were lysed in Homogenization buffer (225 mM mannitol, 75 mM sucrose, and 30 mM Tris-Cl, pH 7.4) using a Dounce homogenizer. The lysate was centrifuged three times at $600 \times g$ for 5 min to remove unlysed cells and nuclei resulting in post-nuclear supernatants (PNS). The cleared lysate was centrifuged at $7000 \times g$ to separate crude mitochondrial pellet and supernatant containing microsomes. The supernatant was cleared by centrifugation at $20,000 \times g$ for 30 min followed by microsome isolation using high-speed centrifugation at $100,000 \times g$ for 1 h. The crude mitochondrial pellet was washed twice in homogenization buffer containing 0.1 mM EGTA at $7000 \times g$ and $10,000 \times g$ for 10 min. MAMs were isolated from crude mitochondria using 30% Percoll gradient centrifugation at $95,000 \times g$ for 1 h in a swinging-bucket rotor. The banded MAM fraction was washed once with phosphate-buffered saline (PBS) before lysing in lysis buffer (50 mM Tris-Cl, pH 7.2, 150 mM NaCl, 2% SDS). The pure mitochondrial fractions were resuspended and washed in mitochondrial resuspension buffer (250 mM mannitol, 0.5 mM EGTA, 5 mM HEPES pH7.4).

Mitochondrial membranes were solubilized using 0.5% (v/v) Digitonin. Protein concentrations for both soluble and pellet fractions were determined by DC protein assay kit (Biorad). For lipidomics of MAMs, the final banded MAM fraction was washed twice with liquid chromatography-mass spectrometry (LC-MS) grade PBS and the final MAM pellet was resuspended in LC-MS grade PBS.

## Cytosolic and nuclear fractionation

Cytosol and nuclear fractions were prepared as previously described[67]. Briefly, cells were seeded a day prior in a 12-well plate. Next day, the cells were collected in PBS and pelleted at $19,000 \times g$ for 15 s. The cell pellet was lysed in 50 mM Tris-Cl, pH 6.8, 150 mM NaCl, 0.1% Nonidet P-40, HALT Protease inhibitors (Pierce) by triturating 5 times with p1000 micropipette. A fraction of this resuspension was collected and labeled as whole-cell lysate. The remainder was centrifuged at $19,000 \times g$ for 15 s and cytosolic supernatant was collected. The nuclear pellet was washed two times with lysis buffer. The whole-cell lysate and nuclear fraction were resuspended in mammalian cell lysis buffer with 1% SDS, sonicated to shear the DNA and solubilize the membranes. Protein concentrations for cytosol, whole cell lysates, and nuclear fractions were determined by DC protein assay kit (Biorad).

## Split luciferase assay to measure ER–mitochondria contacts

Cells seeded a day prior in a 12-well plate were co-transfected with 0.75 μg pcDNA3-Mit-NRluc91 and pcDNA3-CRluc92-ER (kind gift from Jeffrey A. Golden, Brigham and Women's Hospital, Boston) using PEI at a 1:4 (DNA:PEI) ratio, or Lipofectamine 2000 (Invitrogen) as per manufacturer's protocol. Media was changed after 6 and 18 h later, the cells were split into a clear bottom white 96-well plate with 50–100 K cells per well. After 24 h, 30 μM of live-cell substrate Enduren (Promega) was added to cells and incubated for 2–3 h in a 37 °C incubator. The luminescence was measured using a SpectraMax iD3 multi-well plate reader. The luminescence measurements were normalized to the cell viability in each condition. Cell viability was measured using Cell Titer-Glo (Promega) according to the manufacturers' instructions. Relative luminescence units (RLU) for each cell line were normalized to the DMSO treated samples to derive fold changes in RLU. Mean, standard error of means (SEM) and statistical significance were calculated by one way ANOVA with indicated post hoc analysis using GraphPad Prism 5.01 (www.graphpad.com).

## Immunofluorescence and microscopy

HFT stable cell lines, HeLa Kyoto, or COS7 cells were grown on #1.5 cover slips in a 12-well plate and transfected with indicated constructs using Lipofectamine 2000 or siRNAs using RNAiMax. Forty-eight hours post-transfection, cells were washed briefly in PBS and fixed with 4% paraformaldehyde at room temperature for 15 min. Cells were washed in PBS and permeabilized in ice-cold 100% methanol at −20 °C for 10 min. Cells were washed three times in PBS and blocked in 2% bovine serum albumin (BSA) in PBS with 0.3% Triton X-100 for 1 h. The coverslips were incubated overnight with the indicated primary

antibodies in a humidified chamber. The cells were washed and incubated for a further hour with appropriate Alexa-Fluor conjugated secondary antibodies (Molecular Probes) for 1 h in the dark. Cells were washed with PBS, nuclei were stained with Hoechst and mounted on slides. All images were collected using a Nikon A1R scan head with spectral detector and resonant scanners on a Ti-E motorized inverted microscope equipped with ×60 Plan Apo NA 1.4 objective lens. The mitochondrial morphology, Split-GFP, and GFP-MAPPER images were acquired using a Nikon Eclipse Ti2 widefield inverted microscope with CoolLED pE-300 Ultra MB light source and Monochrome Prime BSI Express Scientific CMOS and color DS-Fi3 CMOS camera equipped with ×100 Oil CFI Plan Apo NA 1.45 objective lens. The indicated fluorophores were excited with a 405, 488, or 594 nm laser line. Images were analyzed using FIJI (https://imagej.net/fiji). Using a previously described method[68], co-localized pixel analysis to quantify the ERMCS was performed using an ImageJ macro containing tubeness, colocalization highlighter, and isophotcounter plugins (https://github.com/theramanlab/ganji2022ubxd8). For mitochondrial morphology measurements, we used a previously described ImageJ macro plugin, MiNA (v3.0.1; https://github.com/StuartLab/MiNA)[47].

## Lipid depletion and fatty acid supplementation
Cells were depleted of or supplemented with fatty acids as previously described[69–71]. Briefly, cells were treated with DMEM containing 0.5% lipid-depleted fetal calf serum (LDFCS; S5394, Sigma-Aldrich) for 24 h. 500 mM Oleic acid in DMSO was used as a stock solution to prepare a working solution of 1 mM in DMEM containing 0.5% LDFCS. Cells were treated for 4 h. 500 mM palmitic acid stock solution was prepared in 100% ethanol by heating to 70 °C for 20 min. This stock solution, was used to prepare 0.25 mM palmitic acid solution in DMEM containing 0.5% LDFCS which was heated in a water bath at 50 °C for 2 h. The 0.25 mM palmitic acid solution is cooled down to 37 °C before adding to cells. Cells were incubated in palmitic acid solution for 4 h. All working solutions were prepared immediately prior to treatment.

## Lipid droplet staining
HFT WT or UBXD8 KO cells were grown on #1.5 cover slips in a 12-well plate. Cells were lipid-depleted or supplemented with fatty acids as mentioned above. After treatment, cells were stained for lipid droplets as described previously[72]. Briefly, cells were quickly fixed with 3% paraformaldehyde at room temperature for 20 min. Cells were washed three times in PBS and blocked in 2% BSA in PBS with 0.2% Triton X-100 for 30 min. The coverslips were incubated in dark at room temperature in 2 µM BODIPY 493/503 prepared in 1% BSA in PBS with 0.1% Triton X-100 for 1 h. The cells were washed three times with PBS, nuclei were stained with Hoechst and mounted on slides. All images were collected using a Nikon Eclipse Ti2 widefield inverted microscope with CoolLED pE-300 Ultra MB light source and Monochrome Prime BSI Express Scientific CMOS and color DS-Fi3 CMOS camera equipped with ×100 Oil CFI Plan Apo NA 1.45 objective lens. The indicated fluorophores were excited with either a 405 or 488 nm laser line. Images were analyzed using FIJI (https://imagej.net/fiji). Lipid droplet size and number per cell were calculated using an ImageJ plugin, Aggrecount[73,74] (https://aggrecount.github.io/).

## Real time polymerase chain reaction (PCR)
Equal number of HEK293T WT or UBXD8 KO cells were seeded into a 6-well plate. The next day, total RNA was isolated as per the manufacturer's instructions using the PureLink RNA Mini kit (Thermo fisher). The purified RNA was quantified, and an equal amount of RNA was used for cDNA preparation using iScript cDNA synthesis kit (Biorad). *gapdh* was used as a housekeeping gene was used. Primer sequences used in this study can be found in Supplementary Table 1. Real time PCR was performed using the Powerup SyBr green master mix (Thermo Fisher). Data analyses was carried out using the $2^{-\Delta\Delta Ct}$ method.

## Membrane fluidity measurements
Cells were seeded in clear bottom black 96-well plate and treated with 5 µM CB-5083, lipid depletion or lipid supplementation as described above. Membrane fluidity was measured using a membrane fluidity kit (Abcam, ab189819) as per the manufacturer's instructions. The assay uses a lipophilic, membrane embedding pyrenedecanoic acid probe which undergoes a spectral shift in emission from monomer (Em 400 nm) to excimer (Em 460 nm) based on local membrane fluidity upon excitation at 360 nm. The excimer to monomer (Em 460 nm/Em 400 nm) ratio was calculated for each sample. Then fold changes of ratios (Treated$_{excimer:monomer}$ vs Control$_{excimer:monomer}$) were deduced to provide a relative estimate of membrane fluidity compared to the wildtype untreated control. A fold change <1 indicates ordered membranes relative to control.

## Transmission electron microscopy
Cells were fixed in 2.5% glutaraldehyde, 3% paraformaldehyde with 5% sucrose in 0.1 M sodium cacodylate buffer (pH 7.4), pelleted, and post fixed in 1% $OsO_4$ in veronal-acetate buffer. The cells were stained en bloc overnight with 0.5% uranyl acetate in veronal-acetate buffer (pH 6.0), then dehydrated and embedded in Embed-812 resin. Sections were cut on a Leica EM UC7 ultra microtome with a Diatome diamond knife at a thickness setting of 50 nm, stained with 2% uranyl acetate, and lead citrate. The sections were examined using a FEI Tecnai spirit at 80 KV and photographed with an AMT CCD camera. The images were analyzed manually for the ER-Mitochondrial and ER-PM contacts using FIJI (https://imagej.net/fiji). Briefly, the scale of image was set using Set Scale tool on ImageJ. Followed by measuring the length of ER, PM, or perimeter of mitochondria using freehand line tool. The percent of contact length was determined by taking the ratio of the length of ER, or PM (within contact distances of 25–100 nm) to the perimeter of mitochondria. The data were analyzed using GraphPad Prism 9.4.1 (681) for Windows, GraphPad Software, San Diego, CA, USA (www.graphpad.com).

## TMT-based proteomics
**Sample preparation, digestion, and TMT labeling.** The PNS and MAM fractions were isolated from HEK293T WT or UBXD8 KO cells. One hundred µg of proteins from each sample was precipitated using 15% (v/v) Trichloroacetic acid (TCA) followed by 100% Acetone washes. The protein pellets were resuspended in 200 mM *N*-(2-Hydroxyethyl) piperazine-*N*′-(3-propanesulfonic acid) (EPPS) (pH 8.5) buffer followed by reduction using 5 mM tris(2-carboxyethyl)phosphine (TCEP), alkylation with 14 mM iodoacetamide and quenched using 5 mM dithiothreitol treatments. The reduced and alkylated protein was precipitated using methanol and chloroform. The protein mixture was digested with LysC (Wako) overnight followed by Trypsin (Pierce) digestion for 6 h at 37 °C. The trypsin was inactivated with 30% (v/v) acetonitrile. The digested peptides were labeled with 0.2 mg per reaction of 6-plex TMT reagents (ThermoFisher scientific) (126, 127 N, 127 C, 128 N, 128 C, and 129 N) at room temperature for 1 h. The reaction was quenched using 0.5% (v/v) Hydroxylamine for 15 min. A 2.5-µl aliquot from the labeling reaction was tested for labeling efficiency. TMT-labeled peptides from each sample were pooled together at a 1:1 ratio. The pooled peptide mix was dried under vacuum and resuspended in 5% formic acid for 15 min. The resuspended peptide sample was further purified using C18 solid-phase extraction (SPE) (Sep-Pak, Waters).

## Off-line basic pH reverse-phase (BPRP) fractionation
We fractionated the pooled, labeled peptide sample using BPRP HPLC[75]. We used an Agilent 1200 pump equipped with a degasser and a detector (set at 220 and 280 nm wavelength). Peptides were subjected to a 50-min linear gradient from 5 to 35% acetonitrile in 10 mM ammonium bicarbonate pH 8 at a flow rate of 0.6 ml/min over an

Agilent 300Extend C18 column (3.5 μm particles, 4.6 mm ID and 220 mm in length). The peptide mixture was fractionated into a total of 96 fractions, which were consolidated into 24 super-fractions[76]. Samples were subsequently acidified with 1% formic acid and vacuum centrifuged to near dryness. Each consolidated fraction was desalted via StageTip, dried again via vacuum centrifugation, and reconstituted in 5% acetonitrile, 5% formic acid for LC-MS/MS processing.

### Liquid chromatography and tandem mass spectrometry

Mass spectrometric data were collected on an Orbitrap Lumos mass spectrometer coupled to a Proxeon NanoLC-1000 UHPLC. The 100 μm capillary column was packed with 35 cm of Accucore 150 resin (2.6 μm, 150 Å; ThermoFisher Scientific). The scan sequence began with an MS1 spectrum (Orbitrap analysis, resolution 120,000, 350–1400 Th, automatic gain control (AGC) target $5 \times 10^5$, maximum injection time 50 ms). Data were acquired for 150 min per fraction. SPS-MS3 analysis was used to reduce ion interference[77,78]. MS2 analysis consisted of collision-induced dissociation (CID), quadrupole ion trap analysis, automatic gain control (AGC) $1 \times 10^4$, NCE (normalized collision energy) 35, $q$-value 0.25, maximum injection time 60 ms), isolation window at 0.5 Th. Following acquisition of each MS2 spectrum, we collected an MS3 spectrum in which multiple MS2 fragment ions were captured in the MS3 precursor population using isolation waveforms with multiple frequency notches. MS3 precursors were fragmented by HCD and analyzed using the Orbitrap (NCE 65, AGC $3.0 \times 10^5$, isolation window 1.3 Th, maximum injection time 150 ms, resolution was 50,000). Data are available via ProteomeXchange with identifier PXD039061.

Note on ratio compression. Even though peptide abundance is analyzed in MS3 mode, there is still some interference or ratio compression in these samples. This can happen when (i) a low abundance protein is being analyzed and it has very few peptides (low summed signal), (ii) the peptides are not ionized well (low overall signal), (iii) the protein shares many peptides with other proteins or (iv) true interference from another more signal-dominant peptide that coelutes within the same isolation window, but has a different sequence and originates from a different protein[78,79]. For instance, all the peptides quantified for SREBP1 and 2 were shared thus preventing accurate quantification.

### Data analysis

Spectra were converted to mzXML via MSconvert[80]. Database searching included all entries from the Human UniProt Database (downloaded: August 2018). The database was concatenated with one composed of all protein sequences for that database in the reversed order. Searches were performed using a 50-ppm precursor ion tolerance for total protein level profiling. The product ion tolerance was set to 0.9 Da. These wide mass tolerance windows were chosen to maximize sensitivity in conjunction with Comet searches and linear discriminant analysis[81,82]. TMT tags on lysine residues and peptide N-termini (+229.163 Da for TMT) and carbamidomethylation of cysteine residues (+57.021 Da) were set as static modifications, while oxidation of methionine residues (+15.995 Da) was set as a variable modification. Peptide-spectrum matches (PSMs) were adjusted to a 1% false discovery rate (FDR)[83,84]. PSM filtering was performed using a linear discriminant analysis, as described previously[82] and then assembled further to a final protein-level FDR of 1%[84]. Proteins were quantified by summing reporter ion counts across all matching PSMs, also as described previously[85]. Reporter ion intensities were adjusted to correct for the isotopic impurities of the different TMT reagents according to manufacturer specifications. The signal-to-noise (S/N) measurements of peptides assigned to each protein were summed and these values were normalized so that the sum of the signal for all proteins in each channel was equivalent to account for equal protein loading. Finally, each protein abundance measurement was scaled,

such that the summed signal-to-noise for that protein across all channels equaled 100, thereby generating a relative abundance (RA) measurement.

Downstream data analyses for TMT datasets were carried out using the R statistical package (v4.0.3) and Bioconductor (v3.12; BiocManager 1.30.10). TMT channel intensities were quantile normalized and then the data were log-transformed. The log-transformed data were analyzed with limma-based R package where $P$ values were FDR adjusted using an empirical Bayesian statistical. Differentially expressed proteins were determined using a $\log_2$ (fold change (WT vs UBXD8 KO)) threshold of $>+/-0.65$.

### Gene ontology (GO) functional enrichment analyses of proteomics data

The differentially expressed proteins were further annotated and GO functional enrichment analysis was performed using Metascape online tool (http://metascape.org)[86]. The GO cluster network and protein-protein interaction network generated by metascape and the STRING database (https://string-db.org/), respectively, were imported into Cytoscape software (v3.8.2) to add required attributes (fold changes, $P$ values, gene number, and conditions) and prepared for the visualization. Other proteomic data visualizations were performed using the RStudio software (v1.4.1103), including hrbrthemes (v0.8.0), viridis (v0.6.1), dplyr (v.1.0.7), and ggplot2 (v 3.3.5).

### Lipidomics

**Sample preparation, mass spectrometry, and identification.** For each independent lipidomic experiment, HEK293T WT and UBXD8 KO cells were seeded in triplicate. Two of the three samples for each condition were used for lipidomics (i.e., lipids from duplicate samples were extracted and analyzed in parallel to determine technical variation). The third sample was used to determine the total protein concentration. Cells were washed with PBS, scraped into cold 50% methanol, centrifuged, and the cell pellets were frozen. Next, cells were resuspended in cold 50% methanol and transferred to glass vials. Chloroform was added and the mixture was gently vortexed and centrifuged at $1000 \times g$ for 5 min at 4 °C. Lipids were transferred to a clean glass vial using a glass Hamilton syringe. Lipids were extracted twice using chloroform prior to being dried under nitrogen gas. Samples were normalized according to protein concentration when resuspended in a 1:1:1 solution of methanol:chloroform:isopropanol prior to mass spectrometry (MS) analysis. The samples were stored at 4 °C in an autosampler during data collection.

Lipids were identified and quantitatively measured using ultra high-performance liquid-chromatography high-resolution tandem MS/MS (UHPLC-MS/MS) as recently described[87,88]. Separation of lipids was done by reverse-phase chromatography using a Kinetex 2.6 μm C18 column (Phenomenex 00F-4462-AN) at 60 °C using a Vanquish UHPLC system (Thermo Scientific) and two solvents: solvent A (40:60 water-methanol plus 10 mM ammonium formate and 0.1% formic acid) and solvent B (10:90 methanol-isopropanol plus 10 mM ammonium formate and 0.1% formic acid). UHPLC was performed at a 0.25 ml per min flow rate for 30 min per sample, starting at 25% solvent B and ending at 100% solvent B as described. The column was washed and equilibrated between samples. Samples were run in a semi-random order where WT or UBXD8 KO samples were interspersed with blank samples. Lipids were ionized using a heated electrospray ionization (HESI) source and nitrogen gas and measured using a Q-Exactive Plus mass spectrometer operating at a MS1 resolution of either 70,000 or 140,000 and a MS2 resolution of 35,000. MS1 Spectra were collected over a mass range of 200 to 1,600 m/z with an automatic gain control (AGC) setting of 1e6 and transient times of 250 ms (70,000 resolution) or 520 ms (140,000 resolution). MS2 spectra were collected using a transient time of 120 ms and an AGC setting of 1e5. Each sample was analyzed using negative and positive ion modes. The mass analyzer

was calibrated weekly. SPLASH LIPIDOMIX mass spectrometry standards (Avanti Polar Lipids) were used in determining extraction efficiencies and lipid quantitation. Quality control (QC) samples consisting of lipids extracted from the National Institute of Standards and Technology (NIST) Standard Reference Material 1950 Metabolites in Frozen Human Plasma which contains plasma pooled from 100 healthy donors were used in this study. In parallel to the samples, a control that lacked cells was used to determine any contaminants from the lipid extraction and measurement steps. Any lipids found in the no cell control were removed during analysis steps.

Lipids were identified and quantified using MAVEN[89], EI-MAVEN (Elucidata), Xcalibur (ThermoFisher Scientific), and LipidSearch software (ThermoFisher Scientific). UHPLC retention time, MS[1] peaks, and MS[2] fragments were used to identify lipids. The lipid retention time, MS1 peak shape, isotopic distribution, and MS[2] fragments were visually confirmed for all lipids reported in this study. Peak area was used to determined lipid abundance. Lipids were included if they were observed in 3–6 samples in both UBXD8 KO and WT cells. Missing values in a sample were not imputed. The fold change of each lipid in UBXD8 KO cells relative to its level in WT cells was used to test for statistical difference between UBXD8 KO and WT cells using independent one-tailed $t$ tests and the Benjamini-Hochberg correction method to control for false statistical discovery. The following lipid classes were included in the analysis: cholesteryl esters (CE), diacylglycerol (DG), phosphatidylcholine (PC), phosphatidylethanolamine (PE), phosphatidylglycerol (PG), phosphatidylinositol (PI), phosphatidylserine (PS), and triacylglycerol (TG). Guidelines from the Lipidomic Standards Initiative were followed for lipid species identification and quantification, including consideration of isotopic patterns resulting from naturally occurring $^{13}C$ atoms and isomeric overlap. The following MS[2] information was used to confirm each lipid species: PC fragment of 184.073 (positive mode) and tail identification using formic adduct (negative mode); PE fragment of 196.038 or the tail plus 197.046 (negative mode) and neutral loss (NL) of 141.019 (positive mode); PG fragment of 152.996 plus the identification of the FA tails (negative mode) and NL 189.04 of [M + NH4] + adduct (positive mode); PI fragment of 241.012 (negative) and NL 277.056 of [M + NH4] + adduct (positive mode); PS NL of 87.032 (negative); DG and TG by NL of FA tails (positive mode); and CE fragment of 369.352 or neutral loss of 368.35 (positive).

### Mouse studies

C57BL/6 (stock No.: 000664) and p97[R155H/WT] (B6;129S-Vcptm1Itl/J, Stock No: 021968) were purchased from Jackson Laboratory. p97[R155C/WT] and p97 cKO (VCPFL/FL; CaMKIIa-Cre) were obtained as reported previously[53,54]. All mice utilized in the study were on a C57BL/6 background. Both male and female mice were used in this study and age of mice are mentioned in figure legends. All mice were housed in a pathogen-free environment under controlled environmental conditions with 12 h light-dark cycles at humidity 30–70%, temperature 20–26 °C (68–79 °F) with ventilation sufficient to maintain appropriate temperature and humidity ranges and to control odor, where they received food and water ad libitum until being sacrificed[53].

### Immunohistochemistry of tissue sections

Mice were anesthetized in an isoflurane chamber and perfused with PBS containing heparin. The whole brain was removed from the skull and fixed in 4% PFA overnight at 4 °C, cut coronally into 40 micrometer sections, and stored in cryoprotectant solution at 4 °C for staining. Sections were first rinsed 3 times with TBS and then blocked with blocking solution for 30 min (5% normal goat serum with 0.1% Triton X-100 in TBS). Sections were stained with the primary antibody (SREBP1, 1:500, SCD1: 1:250 dilution) in TBS-0.1% Triton X-100 plus 2% normal goat serum at 4 °C overnight, followed by 3 washes with TBS. Sections were then incubated with the Alexa 488 and 555 tagged

secondary antibodies (1:1000) for 2 h at room temperature, followed by counterstaining with DAPI (1:1000) for 20 min. After three washes with TBS, the sections were mounted on the glass slides. True black (Biotium, NC1125051) was incubated with the sections for 5 min to quench the auto-fluorescence. Slides were cover-slipped using Prolong Gold mounting medium. Images were acquired using a Nikon Eclipse 80i fluorescence microscope. All images were taken with the same fluorescent settings and subsequently adjusted equally for brightness and contrast to ensure accurate pathology quantification. For CA1 regions, ROIs are drawn according to DAPI staining and the mean intensity was measured in each ROI by ImageJ. Background intensities (from three regions per image were subtracted.

### Immunoblot

Mouse cortex was lysed in RIPA buffer with protease inhibitor cocktails (PMSF and PIC) followed by sonication (two cycles of 30 s cycles at 50% power). The protein concentration is estimated by the BCA assay. Samples were loaded into 10% gel and transferred into nitrocellulose membrane. The membranes were blocked by 5% milk in PBS-0.2% Tween20 and incubated with the primary antibody in blocking solution overnight at 4 °C. The membrane was then washed three times with PBS–0.2% Tween20 and incubated with a secondary goat anti-rabbit HRP antibody (1:5000) for 1 h. Blot was rinsed three times with PBS–0.2% Tween20 and probed by a fresh mixture of ECL reagents at dark and then exposed by SYNGENE.

### Statistics and reproducibility

For all experiments, $N \geq 3$ biological replicates for each condition were examined, except for Supplementary Fig. 6g, h ($N = 2$). Representative result from the independent biological replicates are shown. Fold changes, SEM, and statistical analyses were performed using GraphPad Prism version (v5. 01, v9.2.0, v9.3.1, & v9.4.1 (681)) for Windows, GraphPad Software, San Diego, CA, USA (www.graphpad.com). The statistical tests performed, SEM, and statistical significance values are mentioned in the figures and Supplementary Tables.

### Reporting summary

Further information on research design is available in the Nature Portfolio Reporting Summary linked to this article.

### Data availability

The mass spectrometry proteomics data related to Fig. 4 and Supplementary Fig. 4 have been deposited to the ProteomeXchange Consortium via the PRIDE[90] partner repository with the dataset identifier PXD039061. All raw lipidomic data are available in supplementary Dataset 2. The mass spectrometry lipidomics data is available at the NIH Common Fund's National Metabolomics Data Repository (NMDR) website, the Metabolomics Workbench, (https://www.metabolomicsworkbench.org), where it has been assigned Project ID PR001559 [https://doi.org/10.21228/M85X3W] with StudyIDs ST002421 (for whole-cell lipidomics) and ST002422 (MAM fraction lipidomics). This work is supported by NIH grant, U2C-DK119886. Any other data are available from the corresponding author upon request. Source data are provided with this paper.

### Code availability

The ImageJ macro to quantify ER–mitochondria contact sites is made available on GitHub (https://github.com/theramanlab/ganji2022ubxd8).

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

## Acknowledgements

We thank Peter Juo and Karl Munger for critical reading of the manuscript. We are grateful to Jeffrey Golden (Brigham and Women's Hospital, Harvard Medical School) for split luciferase constructs, and the Whitehead Institute Electron Microscopy core for electron microscopy. We are grateful to Christopher Miller (King's College London) for VAPB and PTPIP51 constructs. We are grateful to Marisa Brini and Tito Cali (University of Padova, Italy) for SPLICS constructs. We would also like to thank Jacob Klickstein for developing the ImageJ script for ER–mitochondria co-localization. We thank Brittany Ahlstedt for performing ER stress measurements in wildtype and UBXD8 KO cells. This work is supported by the NIH grant GM127557 to M.R., funds from the University of Arizona Health Sciences, BIO5 Institute and NIH grant AI162671 to J.G.P., NIH grant T32 AG058503 to I.K., and NIH grant AG031867 to C.C.W.

## Author contributions

R.G. and M.R. conceived the studies. R.G performed all experiments except for lipidomics studies (that were performed by Y.X. and J.G.P. and analyzed by Y.X., I.K., and J.G.P.), p97 cKO mice (generated by C.S.C.), and mouse immunohistochemistry and immunoblotting (performed by J.Z.). J.P. and S.P.G. assisted with proteomic studies. M.R. wrote the manuscript with input from R.G. and J.G.P. and C.C.W.

## Competing interests

The authors declare no competing interests.
