## [Peer Review File · Nature Communications]

The p97-UBXD8 complex regulates ER-Mitochondria contact sites by altering membrane lipid saturation and compositionREVIEWER COMMENTS

Reviewer #1 (Remarks to the Author):

In this manuscript, Ganji and colleagues explore the role of p97 and its membrane adaptor UBXD8 in the regulation of ER-mitochondria contacts and lipid homeostasis. Proteomic analysis of membranes enriched in ER-mitochondria contact-sites (MAM fraction) detected the presence of p97 and UBXD8. Using a split-Luciferase reporter to monitor ER-Mitochondria contacts, the authors observed that loss of p97 or UBXD8 function resulted in increased ER-mitochondria contacts. Proteomics and lipidomics analysis showed differential composition of MAMs isolated from UBXD8 KO in comparison to control cells. In particular, changes in lipid metabolic enzymes and increased lipid saturation were observed. The increase in lipid saturation is consistent with measurements performed with probes sensitive to lipid environment. Similarly, suppression of UBXD8 associated phenotypes by supplementation of unsaturated fatty acids are consistent with a role of UBXD8 in controlling phospholipid saturation and membrane fluidity. These findings are in line with previous findings showing a key role of Ubx2, the yeast homologue of UBXD8 in controlling lipid saturation. The authors propose that UBXD8 influences membrane saturation by promoting INSIG1 degradation and/or controlling SCD1 distribution. Finally, the authors examined mouse models expressing disease-associated p97 mutations for the defects observed in tissue culture models.

Understanding the mechanisms underlying organelle dynamics and lipid homeostasis is an important, fast evolving area of research. However, this manuscript, despite some intriguing observations, falls short in advancing our knowledge on the topic. Overall, the manuscript does not follow a coherent storyline and most observations are too preliminary and not sufficiently developed.

Main points:

- The lipid saturation phenotype in UBXD8 is interesting, although not totally unexpected based on the yeast work from Ernst and colleagues. However, the mechanism by which UBXD8 functions in mammalian cells to influence phospholipid unsaturation was not followed up. The authors observe differences in the levels of overexpressed INSIG1 but no data on the endogenous protein is shown. Is this activity related to gp78-regulated degradation of INSIG1? Can INSIG1 depletion restore membrane composition in UBXD8 KO cells? The role of SCD1 is even less clear. The levels of SCD1 are down in the KO cells (by blot) however the TMT MS experiment show only minimal differences.

- The links between p97/UBXD8 to ER-mitochondria contacts are weak and unconvincing. The enrichment of p97 and UBXD8 to the MAM fraction is tiny, if any. These are abundant proteins that are present throughout the ER, including MAMs. Similarly, the changes observed in the proteome of the MAMs from UBXD8 KO cells (Figure 2B) are largely observed in the PNS samples (Supplemental table) suggesting that

they are not specific to the MAMs. In addition, the ER-Mito split-luciferase reporter is poorly characterized. Does the reporter localize to the presumed locations? Are the levels and localization changed upon interfering with p97/UBXD8 function? These controls are essential for any conclusion using this probe. Considering the function of p97/UBXD8 in protein quality control it is reasonable to assume that they influence the behavior of the reporter. Analysis of the probe by microscopy and fractionation experiments would be appropriate.

- The increase in ER-Mito contacts scored by EM is interesting but is this simply because there is an increase in ER surface area in UBXD8? The data presented in Figure 1F suggest that UBXD8 KO cells have more ER. While interesting this observation may be at odds with the reduced processing of SREBP1, critical for lipid synthesis and de novo lipogenesis.

Minor points

- Line 110: "The abundance of 23 proteins was enriched and 28 proteins was depleted in the MAM fraction of UBXD8 KO cells out of a total of 4499 quantified". This sentence does not read well.

- Line 114: "Furthermore, we identified significant enrichment of known p97-UBXD8 substrates such as squalene monooxygenase (SQLE), and HMG-CoA reductase (HMGCR) in UBXD8 KO cells". Please add relevant references

Reviewer #2 (Remarks to the Author):

The ER-mito contact sites are critical for many cellular processes. How these contact sites are remodeled and how they regulate lipid metabolism have drawn great attentions. In the current manuscript, Ganji et al., focused on the role of p97 and its adaptor UBXD8/FAF2 at ER-mito contact sites. They first isolated the MAMs (ER-mitochondria contacts) and found that p97 and its adaptors UBXD8 and UBXD2 were enriched at these contacts, and loss of either p97 or UBXD8 increased the extent of ER-mitochondrial interactions. Proteomic analysis of purified MAMs suggested an enrichment in proteins involved in lipid metabolism, and an increase in saturated lipid species in the UBXD8 KO cells. Defective SREBP1-SCD1 pathway was suggested to be responsible for the elevated contacts in p97-UBXD8 depleted cells and mice with p97 mutations that cause neurodegeneration. Finally, they showed that the abnormal contacts in p97-UBXD8 suppressed cells could be rescued by supplementation with unsaturated fatty acids OA (but not saturated PA) or SCD1 overexpression. These findings are of quite interest to the field of cell biology, although some of the results are preliminary. To improve the manuscript, the authors should address the following points.

Major points:

1. The evidence supporting the up-regulated ER-mitochondrial contacts in p97-UBXD8 depleted cells are not strong. In current manuscript, split luciferase assays, transmission electron microscope (TEM), and colocalized pixels analyses are used by the authors to examine the extent of ER-mitochondrial interactions. None of these assays convincingly showed an increase in ER-mito contacts in p97/UBXD8 depleted cells.

a) While the split luciferase assays may be advantageous over the irreversible split-FP assays for monitoring the dynamic contacts, the cons of this method is that we can NOT see the luciferase signals in cells, in other words, the specificity of these signals were not guaranteed=> Whether these signals from bone fide ER-mitochondrial contacts?

b) The TEM results shown in the manuscript are not convincing, and the labelling on the TEM micrographs make it worse => thin and dotted lines may be better?

c) The authors also used colocalized pixels analyses based on confocal micrographs to examine the level of the contacts. However, the authors did not show raw confocal images of ER-mitochondrial interactions in either control or p97-UBXD8 depleted cells. Only quantifications were shown.

Therefore, the authors should examine ER-mitochondrial interactions more rigorously, at least in some key experiments. For example, using ddFP assays along with ER and mitochondrial markers by live-cell confocal microscope.

2. The major concern for the current work is that the mechanism underlying the regulation of ER-mito contacts by p97-UBXD8 has not been well explored.

1) How does p97-UBXD8 complex regulate Insig1 and further modulate the maturation of SREBP1? How does p97-UBXD8 complex regulate SCD1 level?

2) Furthermore, the finding that the increase in lipid saturation enhanced ER-mito contacts, is quite interesting. However, the mechanisms should be carefully explored, or at least discussed in depth.

a) The author showed that supplementation of mono-unsaturated OA, but not saturated PA, could fully rescue the aberrant ER-mito contacts in p97-UBXD8 depleted cells, which was very interesting. However, it is unclear how OA contributes to rescue the phenotype in the current manuscript. In normal conditions, OA was very likely deposited in LDs in a form of triglycerides. Are LDs involved in the p97/UBXD8-regulation of ER-mito contacts?

b) Supplementation of PA, but not OA, induces stresses on mitochondria (fragmentation) and the Golgi (fragmentation). Whether the difference between OA and PA in rescuing aberrant ER-mito contacts is

due to their difference in toxicity to mitochondria? The author should test if PA (0.25 mM) also caused mito fragmentation, and if yes, at least discuss this possibility in the manuscript.

c) Another major concern is the specificity. As the authors showed in Fig. 4b, not only ER-mito contacts, but ER-plasma membrane contacts, were affected. Whether the p97/UBXD8 complex was also enriched at ER-PM contacts?

d) In cell models, phenotypes of CB-5083 treatment or UBXD8 KO could be reversed by OA. In mouse models, p97 mutations that cause proteinopathies also exhibit increased contacts and display significantly decreased levels of SREBP1 and SCD1. To increase the clinical significance, the authors may treat the animals with OA to test whether the treatment alleviates the neurodegeneration caused by p97 mutations.

3) While p97 and UBXD8 form a complex, whether p97 and UBXD8 function in a complex in regulating ER-mito contacts?

a) The authors showed that p97/UBXD8 were enriched at MAM fractions by cell fractionation, the results should be confirmed by immunofluorescence staining for endogenous p97/UBXD8, or at least overexpression of exogenously tagged p97 or UBXD8 in live-cell imaging => see if p97/UBXD8 were enriched and co-localized at ER-mitochondrial contacts.

b) It will be interesting to see if p97 overexpression could rescue the aberrant ER-mito contacts in UBXD8-KO cells. => p97 and UBXD8 function in the same pathway?

c) Is p97 recruitment to contact sites dependent on UBXD8?

d) It has been reported that association of UBXD8 with the ER-resident rhomboid pseudoprotease UBAC2 specifically restricts trafficking of UBXD8 to cytoplasmic lipid droplets (LDs). UBXD8-mediated recruitment of p97/VCP to LDs increases LD size by inhibiting the activity of adipose triglyceride lipase ATGL. Does UBXD8 directly regulate ER-mito contacts, or affect ER-mito contacts through indirectly regulating lipid composition of other organelles such as LDs?

e) Many p97-mediated processes such as ERAD require its co-factors Ufd1-Npl4. Does p97-UBXD8-mediated regulation of ER-mito contacts require Ufd1-Npl4?

Minor points:

There are 4 main figures and 6 supplementary figures, which is hard to match each other. The authors may consider merging some of the supplementary figures to main figures, making the manuscript easier to follow.

Fig. 1:

1a: The bands of MAM fraction marker FACL4 look odd. Can the authors show more other MAM markers, for example, VAPB, GRP75, and Calnexin?

1e: Can the authors show the TEM micrographs of p97-suppressed cells?

1f: Dots should be shown in the bar graph.

Fig. 2:

2b: Can the authors also show SCD1 and p97 in the plot?

Fig. 3:

3a: Can the authors show the blots of endogenous Insig?

3c: The results showed that the total level of SREBP1 appeared to increase upon p97 suppression, which is different from the UBXD8 KO cell. How the authors reconcile these two results?

3e: As mentioned above, can the authors show more MAM markers?

Fig. 4:

2b: The thinner, dotted lines in the TEM micrograph may be better for visualizing the ER and mito. In addition, can the authors confirm the TEM results with a well-established ER-PM marker MAPPER (24183667)?

4e: The specificity of anti-SREBP1/anti-SCD1 signals in immunofluorescence staining should be carefully tested, for example, by siRNA-mediated suppression.

Fig. S1:

1a: Can the author show more well-established ER-mito contact tethers, for instance VAPB/PTPIP51 and VPS13A?

1d: Scale bar was missing.

1e: Can the authors show the dots in the bar plots?

1f: Did p97 suppression or UBXD8 KO affects the morphology of mitochondria?

Fig. S2:

2b: The quantification for S2b is missing.

Fig. S3 &4

Move to main figures?

Fig. S5

5g: The statistical analysis is missing for the group -SCD1 inhibitor; +OA.

Fig. S6

6b: off topic.

Reviewer #3 (Remarks to the Author):

The AAA-ATPase p97 acts as an ubiquitin-selective unfoldase to control the abundance of several ER and mitochondria proteins involved in protein folding. P97 is recruited to the ER via its adaptors UBXD8 and UBXD2. In their manuscript Ganji et al. uses proteomics, lipidomics and TEM, as well as in vitro and in vivo cell and animal systems, to describe how inhibition of the p97-UBXD8 axis results in increased ER-mitochondria contacts. They find that this is likely due to the inability to activate SREBP1. They further showed that lipid desaturases, such as SCD1, a downstream target of SREBP1, decrease in abundance with UBXD8 KO. The authors find that decreased SCD1 expression is linked to decreased production of specific lipid species, including Oleic acid, which can be supplemented to rescue the increase in ER-mitochondria contact sites, likely due to restoration of lipid fluidity. Finally, the authors performed in vivo mouse studies to validate the clinical relevance of their findings to neurodegenerative diseases.

Major Concerns:

1. While global membrane fluidity and lipid alterations were measured for figures 2 and 4, this does not indicate that lipid composition and fluidity are altered specifically at MAMs. To indicate specificity to MAMs, the authors could perform lipidomics on isolated membranes, as well as use a MAM marker to measure membrane fluidity specific to those sites.

2. The authors compare the proteomes of isolated MAMs taken from WT and UBXD8 KO cells through use of TMT-MS. Through this analysis they only identify a handful of proteins being significantly altered. The authors initially note a significance cutoff of $abs(\log_2(WT/KO)) > 1$; however, the only pathways noted that were not expected (either due to their roles in contacts or p97-UBXD8 pathway) are below this threshold. In following experiments, the authors identify several other proteins and processes that change in abundance in UBXD8 KO cells (SREBP1, SCD1 e.g.). Why were these proteins not identified in

the TMT analysis? The knowledge added by the TMT study is not evident. Additionally, there is very little follow-up from this experiment, as most motivations for subsequent figures are from previous data.

3. Immunofluorescence microscopy imaging should be used to validate the western blots indicating protein localization to different fractions. Additionally, the authors suggest that the observed lipid alterations with UBXD8 KO are due to the inability of SREBP1 to translocate, which would also be easily tested via immunofluorescence microscopy.

4. The authors systematically identify important components downstream of UBXD8 throughout the paper. Inclusion of a model figure putting all the findings together into a cartoon would aid the interpretation and emphasize the pathways that are manipulated.

Minor Concerns:

1. For figure 3d, why are all the graphs normalized to the control except for mature SREBP1?
2. In figure 3f SREBP1 and SCD1 abundances are mildly perturbed by UBXD8 KO, is this significant?
3. Addition of quantification for SCD1 abundance from the blot shown in supplementary figure 5f would make interpretation easier.
4. Indicate the distance between split luciferase probes necessary for signal and how that distance compares to that found for ERMCS. Additionally, provide information on how these probes are localized to their respective organelles.
5. As validation of increased ERMCSs with UBXD8 the authors indicate that there is no significant change in mitochondria abundance; however, they do not mention whether there also are ER abundance or morphology changes.
6. The sentence in line 631-632 has a typo.
7. Referring to ER-mitochondria contact sites as simply “contacts” is a confusing shorthand. As an alternative, an abbreviation (e.g., ERMCS) could be considered and would be more accurate.
8. It would be good not to modulate twice in the title.

Reviewer #4 (Remarks to the Author):

In this study, Ganji et al. utilise a variety of approaches in HEK293 cells to demonstrate that loss of the p97 AAA ATPase and its adaptor UBXD8 leads to decreased SREBP1 activation, decreased desaturase activity and membrane fluidity and this leads to increased ER-mito contact sites. Importantly, they show that this can be rescued by oleic acid supplementation or SCD1 overexpression. In addition they show that there is also increased mito-ER contact sites that can be rescued with oleic acid in cells with the IBMFPD common mutation p97 R155H, and that SCD1 and active SREBP1 levels are decreased in the brains of mouse models with conditional knockout of p97 or p97R155C/WT . Overall, these are interesting findings relating p97 to SREBP1 driven changes in lipid saturation that impact ER-mito contact sites.

My main comment is centered around the presentation and interpretation of the lipidomics data. The authors analyse a range of phospholipids, DAGs, TAGs and cholesterol esters and present the data as graphs with the data plotted as scatter charts of the log₁₀ peak areas of the individual lipid species in the KO versus the WT or as heatmaps and dot plots showing fold change.

1) P values of lipids that change are shown in supplementary tables but it would be helpful for the reader if an indication of significance or variability in the data was also incorporated into the figures. For example, a volcano plot of KO versus WT lipid species. In addition, are all the species shown in the heat maps in figure s4 d and e significantly different or was any p value or q value cut-off applied to select which lipids to show in the heatmap?

2) The authors argue that the lipid species that are most increased in the KO cells have 1 or less double bonds in the acyl chain. They then go on to demonstrate the importance of SCD and how monounsaturated or fatty acids with one double bond can rescue the phenotype. In addition, from the dot plots showing the two tails of PC, lipids containing monounsaturated fatty acids seem increased in the KO. If the aim of the lipidomic data is to highlight that there is more saturated lipids, it doesn't seem to show this (for example, the biggest fold change in PE lipids contain 46:6, 40:5, 40:6). Based on the other data in the manuscript, I don't doubt that there seems to be an increase in saturation but the lipidomic data as currently presented doesn't support this. As the data presented here shows relative abundance of individual species between cell types but not the absolute abundance of different lipid species, are the lipid species containing saturated fatty acids more abundant in the cell on a molar basis? Could the authors look at total fatty acid abundance and the ratio of C16:0/C16:1, C18:0/C18:1 to get a better quantitative indication of saturation?

3) Is there a mistake in s4f as it seems to contain two identical dot plots?

Does UBXD8 loss impact mitochondria function such as respiration, and does oleic acid rescue this?

Response to Review

We thank the reviewers for their detailed, constructive comments. We have addressed virtually all of the the comments and added substantial new data that has significantly improved the manuscript. However, some comments would have required extensive experimentation that would not significantly alter the conceptual message of the paper. We hope the reviewers agree that the experiments presented represent a compelling study that makes conceptually novel contributions to our understanding of how ER-mitochondria contact sites (ERMCS) are regulated by membrane lipid composition. The reviews are reproduced in their entirety (bold), our responses are in *blue*.

Reviewer#1

In this manuscript, Ganji and colleagues explore the role of p97 and its membrane adaptor UBXD8 in the regulation of ER-mitochondria contacts and lipid homeostasis. Proteomic analysis of membranes enriched in ER-mitochondria contact-sites (MAM fraction) detected the presence of p97 and UBXD8. Using a split-Luciferase reporter to monitor ER-Mitochondria contacts, the authors observed that loss of p97 or UBXD8 function resulted in increased ER-mitochondria contacts. Proteomics and lipidomics analysis showed differential composition of MAMs isolated from UBXD8 KO in comparison to control cells. In particular, changes in lipid metabolic enzymes and increased lipid saturation were observed. The increase in lipid saturation is consistent with measurements performed with probes sensitive to lipid environment. Similarly, suppression of UBXD8 associated phenotypes by supplementation of unsaturated fatty acids are consistent with a role of UBXD8 in controlling phospholipid saturation and membrane fluidity. These findings are in line with previous findings showing a key role of Ubx2, the yeast homologue of UBXD8 in controlling lipid saturation. The authors propose that UBXD8 influences membrane saturation by promoting INSIG1 degradation and/or controlling SCD1 distribution. Finally, the authors examined mouse models expressing disease-associated p97 mutations for the defects observed in tissue culture models. Understanding the mechanisms underlying organelle dynamics and lipid homeostasis is an important, fast evolving area of research. However, this manuscript, despite some intriguing observations, falls short in advancing our knowledge on the topic. Overall, the manuscript does not follow a coherent storyline and most observations are too preliminary and not sufficiently developed.

Main points:

The lipid saturation phenotype in UBXD8 is interesting, although not totally unexpected based on the yeast work from Ernst and colleagues. However, the mechanism by which UBXD8 functions in mammalian cells to influence phospholipid unsaturation was not followed up. The authors observe differences in the levels of overexpressed INSIG1 but no data on the endogenous protein is shown. Is this activity related to gp78-regulated degradation of INSIG1? Can INSIG1 depletion restore membrane composition in UBXD8 KO cells? The role of SCD1 is even less clear. The levels of SCD1 are down in the KO cells (by blot) however the TMT MS experiment show only minimal differences.

We were unable to probe for endogenous INSIG1 due to difficulties in detecting the endogenous protein with multiple commercial antibodies (including Proteintech, Abcam, and Cell Signaling Technology). This may be in part due its reported short life half-life [PMID: 17043353]. Since we could not consistently detect endogenous INSIG1 we employed over-expression studies using INSIG1-HA/FLAG. Our data with overexpressed INSIG1 agree with many published reports on the known role of p97-UBXD8 in regulating INSIG1 degradation [PMID: 18835813; PMID: 21115839]. Given the lack of reliable antibodies and the significant amount of data we provide to

support the role of the SREBP1 pathway in regulating ERMCS (see below), we felt that the INSIG1 rescue data would not add substantially to our findings and we elected not to perform this study.

We now provide new data with a validated gp78 KO cell line generated by CRISPR editing [PMID: 29275994]. We show in Fig 6e that the degradation of INSIG1 is dependent on GP78 as expected from previous reports [PMID: 17043353; PMID: 23087214]. Furthermore, we also observe an increase in ERMCS (Fig 2d) and decrease in SCD1 protein levels validating this pathway (Fig 6e).

We agree with the reviewer that the TMT experiment shows only minimal differences in SCD1 protein levels in UBXD8 KO cells. This may be due to a number of reasons. The MAM purifications are not pure and there is always some organelle contamination that varies from experiment to experiment. Furthermore, ratio compression in TMT studies can repress fold changes. Even though peptide abundance is analyzed in MS3 mode, there is still some interference/compression. This can happen when (i) a low abundance protein is being analyzed and it has very few peptides (low summed signal), (ii) the peptides are not ionized well (low overall signal), (iii) the protein shares many peptides with other proteins or (iv) true interference from another more signal-dominant peptide that coelutes within the same isolation window, but has a different sequence and originates from a different protein. For instance, all the peptides quantified for SREBP1 and 2 were shared thus preventing accurate quantification. However, our proteomic studies also identified desaturases FADS1 (\log_2FC WT:UBXD8 KO = 0.60) and FADS2 (\log_2FC WT:UBXD8 KO = 0.42) to be depleted modestly in UBXD8 KO (Fig. 4b. Supplementary Table 1). For clarity, we now include a discussion of this issue in the Methods section.

We have provided multiple lines of evidence showing that defective ERMCS levels in UBXD8 KO cells is through SREBP1-SCD1 pathway using complementary approaches: 1) The ERMCS defect in UBXD8 or p97 depleted cells can be rescued with over-expression of the mature form of SREBP1 (Supplementary Fig 6f). 2) Defective ERMCS levels in UBXD8 KO or p97 depleted cells can be rescued with over-expression of wildtype but not with the catalytically dead SCD1 (Fig 6i & Supplementary Fig 6j). 3) ERMCS increase upon pharmacological inhibition of SCD1 (Supplementary Fig 6h). 4) The defective ERMCS phenotype in UBXD8 KO or p97 inhibited cells can be rescued with unsaturated oleic acid but not saturated palmitic acid (Fig 6j). 5) We also provide new lipidomics data of MAM fractions and show that MAMs from UBXD8 KO cells have a significant increase in phospholipids with saturated tails and a decrease in phospholipids with unsaturated tails (Fig 5b). 6) We further strengthen this model by providing new data using gp78 KO cells to show (i) an increase in ERMCS, (ii) decreased SCD1 and (iii) inactivated SREBP1.

The links between p97/UBXD8 to ER-mitochondria contacts are weak and unconvincing. The enriched of p97 and UBXD8 to the MAM fraction is tiny, if any. These are abundant proteins that are present throughout the ER, including MAMs. Similarly, the changes observed in the proteome of the MAMs from UBXD8 KO cells (Figure 2B) are largely observed in the PNS samples (Supplemental table) suggesting that they are not specific to the MAMs.

We have now provided quantification data showing that UBXD8 is clearly enriched at MAMs (Fig 1a-d). In the case of p97, we do not see an enrichment of p97 at MAMs compared to PNS but we do see a modest but significant decrease in recruitment of p97 at the MAMs in UBXD8 KO compared to WT (Supplementary Fig 1c-d).

We agree that many MAM proteins are also found in the PNS proteome samples, but as the reviewer pointed out, these proteins are present throughout the ER and enriched at MAMs. To show the enrichment of known MAM proteins relative to PNS, we have compared the peptide

numbers identified in PNS and MAM samples. As shown in Fig 4c we find that many MAM proteins are significantly enriched. To further show this enrichment, we have compared the peptide numbers identified for well-established ERMCS proteins across the MAM and PNS samples (Fig 4c right panel).

In addition, the ER-Mito split-luciferase reporter is poorly characterized. Does the reporter localize to the presumed locations? Are the levels and localization changed upon interfering with p97/UBXD8 function? These controls are essential for any conclusion using this probe. Considering the function of p97/UBXD8 in protein quality control it is reasonable to assume that they influence the behavior of the reporter. Analysis of the probe by microscopy and fractionation experiments would be appropriate.

The split-luciferase reporter system was generated in the Golden lab. They validated the constructs by showing the localization of the split reporters to the respective organelles and with the positive control, REEP1, a known ERMCS tether protein [PMID: 26201691; PMID: 28760823]. However, to address the reviewer's concern, we quantified the levels of the split luciferase proteins by immunoblotting. We did not see any significant changes in the expression levels of these constructs in cells depleted of p97 or UBXD8 (Supplementary Fig 2c-d). Further, we used an additional established ERMCS tether complex, VAPB-PTPIP51 to quantify ERMCS using this reporter. We found that ERMCS increased upon the over-expression of VAPB and PTPIP51 as previously reported [PMID: 24893131]. Thus, the split luciferase system is a reliable reporter of contacts and the increased ERMCS in p97-UBXD8 loss of function cells is not due to inappropriate stabilization of these reporters.

Additionally, we now provide new data where we used a well-accepted fluorescent protein based split system (Split-GFP) targeted to ERMCS. The split-GFP reporter is also increased upon UBXD8 knockdown or p97 inhibition (Fig 3a-b) and provides an additional visual assay to measure contacts.

The increase in ER-Mito contacts scored by EM is interesting but is this simply because there is an increase in ER surface area in UBXD8? The data presented in Figure 1F suggest that UBXD8 KO cells have more ER. While interesting this observation may be at odds with the reduced processing of SREBP1, critical for lipid synthesis and de novo lipogenesis.

We thank the reviewer for this comment. We do not find any significant defects in mitochondria number or morphology in UBXD8 KO cells. We have now measured the ER length in the TEM micrographs (Fig 3e). We do not see any significant changes in the ER length between WT or UBXD8 KO cells. This suggests that the defects in the SREBP1 processing in UBXD8 KO cells is not due to reduced ER area or changes in ER morphology.

Minor points

- Line 110: "The abundance of 23 proteins was enriched and 28 proteins was depleted in the MAM fraction of UBXD8 KO cells out of a total of 4499 quantified". This sentence does not read well.

We have now rephrased the sentence for clarity.

- Line 114: "Furthermore, we identified significant enrichment of known p97-UBXD8 substrates such as squalene monooxygenase (SQLE), and HMG-CoA reductase (HMGCR) in UBXD8 KO cells". Please add relevant references.

We have now added the relevant references in the revised manuscript.

Reviewer#2

The ER-mito contact sites are critical for many cellular processes. How these contact sites are remodeled and how they regulate lipid metabolism have drawn great attentions. In the current manuscript, Ganji et al., focused on the role of p97 and its adaptor UBXD8/FAF2 at ER-mito contact sites. They first isolated the MAMs (ER-mitochondria contacts) and found that p97 and its adaptors UBXD8 and UBXD2 were enriched at these contacts, and loss of either p97 or UBXD8 increased the extent of ER-mitochondrial interactions. Proteomic analysis of purified MAMs suggested an enrichment in proteins involved in lipid metabolism, and an increase in saturated lipid species in the UBXD8 KO cells. Defective SREBP1-SCD1 pathway was suggested to be responsible for the elevated contacts in p97-UBXD8 depleted cells and mice with p97 mutations that cause neurodegeneration. Finally, they showed that the abnormal contacts in p97-UBXD8 suppressed cells could be rescued by supplementation with unsaturated fatty acids OA (but not saturated PA) or SCD1 overexpression. These findings are of quite interest to the field of cell biology, although some of the results are preliminary. To improve the manuscript, the authors should address the following points.

Major points:

1. The evidence supporting the up-regulated ER-mitochondrial contacts in p97-UBXD8 depleted cells are not strong. In current manuscript, split luciferase assays, transmission electron microscope (TEM), and colocalized pixels analyses are used by the authors to examine the extent of ER-mitochondrial interactions. None of these assays convincingly showed an increase in ER-mito contacts in p97/UBXD8 depleted cells.

a) While the split luciferase assays may be advantageous over the irreversible split-FP assays for monitoring the dynamic contacts, the cons of this method is that we can NOT see the luciferase signals in cells, in other words, the specificity of these signals were not guaranteed=> Whether these signals from bone fide ER-mitochondrial contacts? Therefore, the authors should examine ER-mitochondrial interactions more rigorously, at least in some key experiments. For example, using ddFP assays along with ER and mitochondrial markers by live-cell confocal microscope.

We agree with the reviewer that the split luciferase assays do not allow us to visualize the contacts. As requested, we have now performed the ERMCS quantifications using well established split-GFP constructs that are targeted to ERMCS. The results from split-GFP assay agree with our split-luciferase assays showing increased GFP puncta (ERMCS) upon UBXD8 knock down or p97 inhibition with CB5083 (Fig 3e).

To further address the specificity of the split luciferase reporter, we quantified the levels of the split luciferase proteins by immunoblotting. We did not see any significant changes in the expression levels of these constructs in cells depleted of p97 or UBXD8 (Supplementary Fig 2c-d). Further, we used an additional established ERMCS tether complex, VAPB-PTPIP51 to quantify ERMCS. We found that ERMCS (measured using the split luciferase reporter) increased upon the over-expression of VAPB and PTPIP51 as previously reported [PMID: 24893131]. Thus, the previously validated split luciferase system is a reliable reporter of contacts and the increased ERMCS in p97-UBXD8 loss of function cells is not due to inappropriate stabilization of these reporters.

b) The TEM results shown in the manuscript are not convincing, and the labelling on the TEM micrographs make it worse => thin and dotted lines may be better?

We have now replaced the thick lines with thin dotted lines in TEM images

c) The authors also used colocalized pixels analyses based on confocal micrographs to examine the level of the contacts. However, the authors did not show raw confocal images of ER-mitochondrial interactions in either control or p97-UBXD8 depleted cells. Only quantifications were shown.

We have now provided the representative raw confocal images (Supplementary Fig 3b).

2. The major concern for the current work is that the mechanism underlying the regulation of ER-mito contacts by p97-UBXD8 has not been well explored.

1) How does p97-UBXD8 complex regulate Insig1 and further modulate the maturation of SREBP1? How does p97-UBXD8 complex regulate SCD1 level?

We apologize that we did not elaborate on how p97-UBXD8 regulate contacts through the SREBP1-SCD1 pathway in the Discussion. We have now included a detailed discussion and model (Fig 8f) on how our data supports a role for p97-UBXD8 in modulating ERMCS by regulating membrane lipid saturation.

We have provided multiple lines of evidence showing that defective ERMCS levels in UBXD8 KO is through SREBP1-SCD1 pathway: 1) The ERMCS defect in UBXD8 or p97 depleted cells can be rescued with over-expression of the mature form of SREBP1 (Supplementary Fig 6f). 2) Defective ERMCS levels in UBXD8 KO or p97 depleted cells can be rescued with over-expression of wildtype but not with catalytically dead SCD1 (Fig 6i & Supplementary Fig 6j). 3) ERMCS increase upon pharmacological inhibition of SCD1 (Supplementary Fig 6h). 4) The defective ERMCS phenotype in UBXD8 KO or p97 inhibited cells could be rescued with unsaturated oleic acid but not with palmitic acid (Fig 6j). 5) We also provide new lipidomics data of MAM fractions and show that MAMs from UBXD8 KO cells have a significant increase in phospholipids with saturated tails and a decrease in phospholipids with unsaturated tails (Fig 5b). 6) We further strengthen this model by providing new data using gp78 KO cells to show (i) an increase in ERMCS, (ii) decreased SCD1 and (iii) inactivated SREBP1. 7) Finally, we provide new data where we show that the inactivation of SREBP1 in UBXD8 KO cells occurs primarily in the MAM fractions even though SREBP1 is present throughout the ER (Fig 6g-h).

Taken together, we propose that p97-UBXD8 regulate the levels of ERMCS by regulating activation of SREBP1 at ERMCS which in turn modulates membrane lipid saturation and composition through SCD1. A recent pre-print from the Lippincott-Schwarz group using high speed molecular tracking of the ERMCS tether VAPB found that VAPB diffused rapidly in and out of contact sites, but that VAPB molecules within contacts displayed significantly decreased diffusion within contacts relative to the surrounding ER (Obara et al., 2022, <https://doi.org/10.1101/2022.09.03.505525>). While the mechanisms that regulate limited mobility within ERMCS are not known, our studies suggest that lipid order may have a significant impact on the diffusion rates of tethers.

2) Furthermore, the finding that the increase in lipid saturation enhanced ER-mito contacts, is quite interesting. However, the mechanisms should be carefully explored, or at least discussed in depth.

We appreciate the reviewer for this comment. As stated above, we provide new lipidomics data of MAM fractions and show that MAMs from UBXD8 KO cells have a significant increase in phospholipids with saturated tails and a decrease in phospholipids with unsaturated tails indicating increased membrane saturation (Fig 5b). We also provide new data where we show that the inactivation of SREBP1 in UBXD8 KO cells occurs primarily in the MAM fractions even though SREBP1 is present throughout the ER (Fig 6g-h). These results suggest SREBP activation occurs at ERMCS and the resulting loss of SCD1 increases membrane saturation at ERMCS. A detailed discussion and model are now included.

a) The author showed that supplementation of mono-unsaturated OA, but not saturated PA, could fully rescue the aberrant ER-mito contacts in p97-UBXD8 depleted cells, which was very interesting. However, it is unclear how OA contributes to rescue the phenotype in the current manuscript. In normal conditions, OA was very likely deposited in LDs in a form of triglycerides. Are LDs involved in the p97/UBXD8-regulation of ER-mito contacts?

We provide new data where we measured LDs in WT and UBXD8 KO with or without lipid supplementation (Supplementary Fig 7a-c). Previous studies reported that p97 and UBXD8 inhibit the lipolytic lipase, ATGL, causing an increase in LD size. These studies were performed by over-expressing UBXD8 and depletion studies were not reported [PMID: 23297223]. Corroborating these earlier studies, we found that LDs in UBXD8 KO cells were smaller in size than WT cells. Importantly, the number of LDs between WT and UBXD8 KO cells was comparable in all the tested conditions suggesting that the LD numbers were not defective in UBXD8 KO. While supplementation with OA in UBXD8 KO rescued the ERMCS phenotype, we believe that this is not due to differential regulation of LDs in wildtype and UBXD8 KO cells. We come to this conclusion as we observe equivalent increases in LD size in both wildtype and UBXD8 KO cells. Therefore, we conclude that the potential role of p97-UBXD8 on ERMCS is a direct effect and not by regulating LDs.

b) Supplementation of PA, but not OA, induces stresses on mitochondria (fragmentation) and the Golgi (fragmentation). Whether the difference between OA and PA in rescuing aberrant ER-mito contacts is due to their difference in toxicity to mitochondria? The author should test if PA (0.25 mM) also caused mito fragmentation, and if yes, at least discuss this possibility in the manuscript.

We assessed mitochondrial network morphology and area upon OA or PA supplementation using fluorescence microscopy and a published image analysis script mitochondrial network analysis (MINA) [PMID: 28314612]. Antimycin and oligomycin (AO) co-treatment was used as a positive control in wildtype cells to collapse membrane potential and cause fragmentation (Supplementary Fig. 7d-g). Unlike AO treatment which caused significant mitochondrial fragmentation, OA and PA treatment induced very minor effects under the conditions tested and the changes were comparable between wildtype and UBXD8 KO cells (Supplementary Fig. 7d-g).

Additionally, we performed split-luciferase assays in HEK293T WT cells with Brefeldin A (Supplementary Fig 2g), which disrupts the Golgi. We did not see any change in the ERMCS levels upon Brefeldin A treatment. Thus, ERMCS are not impacted by the fragmentation state of the Golgi apparatus.

c) Another major concern is the specificity. As the authors showed in Fig. 4b, not only ER-

mito contacts, but ER-plasma membrane contacts, were affected. Whether the p97/UBXD8 complex was also enriched at ER-PM contacts?

The p97/UBXD8 effect on the membrane fluidity is via inactivation of SREBP1 thereby decreasing the levels of the desaturase, SCD1. We think this occurs at ERMCS because our subcellular fractionation of MAMs shows that UBXD8/p97, SREBP1, and SCD1 are enriched at ERMCS (Fig 6f-h). Furthermore, in new data we show that SREBP1 activation occurs at ERMCS relative to the rest of the ER (Fig 6f-h). However, changes in SCD1 impact global membrane fluidity which is why we observe increased ER-PM contacts. Due to the significant number of new studies, we performed, we felt these time intensive fractionation studies did not add appreciably to the manuscript and we elected not to perform them. However, we have bolstered our ER-PM quantification using another ER-PM reporter (GFP-MAPPER). In agreement with our TEM analysis, we found that UBXD8 or p97 knockdown as well as p97 inhibition (by CB5083), caused an increase in ER-PM contacts (Fig 7c-d).

d) In cell models, phenotypes of CB-5083 treatment or UBXD8 KO could be reversed by OA. In mouse models, p97 mutations that cause proteinopathies also exhibit increased contacts and display significantly decreased levels of SREBP1 and SCD1. To increase the clinical significance, the authors may treat the animals with OA to test whether the treatment alleviates the neurodegeneration caused by p97 mutations.

The reviewer's comment is well-taken and this is an area we are very interested in exploring. However, a number of issues prevented us from completing this study in a timely manner. There is significant optimization that needs to be performed to determine dose and route of administration of OA. Furthermore, the p97-R155C and the conditional KO mice take at least 12-13 months to develop disease phenotypes, we think these experiments are beyond the scope of current study and hence elected to not perform these experiments. Instead, in the Discussion, we have referenced two recent studies which reported that a lipid-controlled diet or modulation of lipid biosynthesis may have therapeutic implications in mouse models of p97 proteinopathy [PMID: 33410456; PMID: 24158850]. These studies show that feeding with a lipid-enriched diet results in improved survival, motor activity, muscle pathology and autophagy in mutant mice.

3) While p97 and UBXD8 form a complex, whether p97 and UBXD8 function in a complex in regulating ER-mito contacts?

We thank the reviewer for this suggestion. We have already shown that over-expressing different domain mutants of UBXD8 in UBXD8-depleted cells cannot rescue defective ERMCS. To the reviewers' comment, a UBX domain mutant in UBXD8 that cannot bind p97 did not rescue ERMCS (Fig 2b & Supplementary Fig 2e) suggesting a p97-UBXD8 complex is required in this process. Furthermore, we now provide new data that shows p97 localization to ERMCS is diminished in UBXD8 KO cells (Supplementary Fig 1c-d). Finally, we now also show that over-expression of p97 in UBXD8 depleted conditions cannot rescue ERMCS to wildtype levels (Supplementary Fig 2f) indicating that UBXD8 and p97 function as a complex to regulate ERMCS.

a) The authors showed that p97/UBXD8 were enriched at MAM fractions by cell fractionation, the results should be confirmed by immunofluorescence staining for endogenous p97/UBXD8, or at least overexpression of exogenously tagged p97 or UBXD8 in live-cell imaging => see if p97/UBXD8 were enriched and co-localized at ER-mitochondrial contacts.

We now show enrichment of UBXD8 to ERMCS using microscopy. We imaged Cos-7 cells transiently transfected with mito-BFP, SEC61 β and mCherry-UBXD8 to observe UBXD8 localization at ERMCS. We found that relative to ER marker, mCherry-UBXD8 was enriched at ERMCS (Fig. 1c, d). We were unable to show similar microscopy-based p97 enrichment at ERMCS at this resolution due to very high cellular abundance of p97. In any case, our fractionation studies do not show enrichment of p97 at MAMs. However, we provide new data to show that p97 localization to MAMs is diminished in UBXD8 KO cells (Supplementary Fig 1c-d).

b) It will be interesting to see if p97 overexpression could rescue the aberrant ER-mito contacts in UBXD8-KO cells. => p97 and UBXD8 function in the same pathway?

We provide new data where we over-express p97 in UBXD8 KO cells. We did not observe rescue under these conditions, suggesting that UBXD8 and p97 function as a complex in the same pathway to regulate ERMCS.

c) Is p97 recruitment to contact sites dependent on UBXD8?

We thank the reviewer for this suggestion. Using subcellular fractionation, we provide new data to show that p97 localization to MAMs is diminished in UBXD8 KO cells (Supplementary Fig 1c-d).

d) It has been reported that association of UBXD8 with the ER-resident rhomboid pseudoprotease UBAC2 specifically restricts trafficking of UBXD8 to cytoplasmic lipid droplets (LDs). UBXD8-mediated recruitment of p97/VCP to LDs increases LD size by inhibiting the activity of adipose triglyceride lipase ATGL. Does UBXD8 directly regulate ER-mito contacts, or affect ER-mito contacts through indirectly regulating lipid composition of other organelles such as LDs?

We provide new data where we measured LDs in WT and UBXD8 KO with or without lipid supplementation (Supplementary Fig 7a-c). Previous studies reported that p97 and UBXD8 inhibit the lipolytic lipase, ATGL, causing an increase in LD size. These studies were performed by over-expressing UBXD8 and depletion studies were not reported [PMID: 23297223]. Corroborating these earlier studies, we found that LDs in UBXD8 KO cells were smaller in size than WT cells. Importantly, the number of LDs between WT and UBXD8 KO cells was comparable in all the tested conditions suggesting that the LD numbers were not defective in UBXD8 KO. While supplementation with OA in UBXD8 KO rescued the ERMCS phenotype, we believe that this is not due to differential regulation of LDs in wildtype and UBXD8 KO cells. We come to this conclusion as we observe equivalent increases in LD size in both wildtype and UBXD8 KO cells. Therefore, we conclude that the potential role of p97-UBXD8 on ERMCS is a direct effect and not by regulating LDs.

e) Many p97-mediated processes such as ERAD require its co-factors Ufd1-Npl4. Does p97-UBXD8-mediated regulation of ER-mito contacts require Ufd1-Npl4?

We have performed ERMCS quantification upon siRNA-mediated knockdown of UFD1 or NPL4. However, we did not see any impact on ERMCS, which rules out a role for UFD1-NPL4 in this process (Fig 2c). Furthermore, in previous reports INSIG1 degradation has only been attributed to UBXD8 and not to UFD1-NPL4.

Minor points:

There are 4 main figures and 6 supplementary figures, which is hard to match each other. The authors may consider merging some of the supplementary figures to main figures, making the manuscript easier to follow.

We have revised the manuscript to now contain 8 main figures with 8 associated supplementary figures.

Fig. 1: 1a: The bands of MAM fraction marker FACL4 look odd. Can the authors show more other MAM markers, for example, VAPB, GRP75, and Calnexin?

We have now probed our MAM fractions with other MAM markers including VAPB, SIGMA1R, and Calnexin as per reviewer suggestion (Fig 1a, Supplementary Fig 1c).

1e: Can the authors show the TEM micrographs of p97-suppressed cells?

A previous study by Edward Fon's group has shown that inhibiting p97 using the allosteric p97 inhibitor NMS873 increases ERMCS by TEM. This due to the inability to degrade the MFN1/2 ERMCS tether [PMID: 29676259]. Given that our studies corroborate and complement their findings, we decided that it was not the best use of funds and time to re-produce the exact study. We have included this reference in the manuscript.

1f: Dots should be shown in the bar graph.

We apologize for the mistake. We have included the individual data points in the bar graph.

Fig.2: 2b: Can the authors also show SCD1 and p97 in the plot?

We have now shown the data points corresponding to SCD1, SREBP1, SREBP2, FADS1, FADS2, and p97 in the Volcano plot (Fig 4b). While our immunoblotting studies find significant differences in SREBP and SCD1 protein levels, the same phenotype is not apparent in the TMT proteomics. This is likely due ratio compression in TMT studies. Even though peptide abundance is analyzed in MS3 mode, there is still some interference/compression. This can happen when (i) a low abundance protein is being analyzed and it has very few peptides (low summed signal), (ii) the peptides are not ionized well (low overall signal), (iii) the protein shares many peptides with other proteins or (iv) true interference from another more signal-dominant peptide that coelutes within the same isolation window, but has a different sequence and originates from a different protein. For instance, all the peptides quantified for SREBP1 and 2 were shared thus preventing accurate quantification. However, our proteomic studies also identified desaturases FADS1 (\log_2FC WT:UBXD8 KO = 0.60) and FADS2 (\log_2FC WT:UBXD8 KO = 0.42) to be depleted modestly in UBXD8 KO (Fig. 4b. Supplementary Table 1). For clarity we now include a discussion of this issue in the Methods section.

Fig. 3:3a: Can the authors show the blots of endogenous Insig?

We were unable to probe for endogenous INSIG1 due to difficulties in detecting the endogenous protein with multiple commercial antibodies (including Proteintech, Abcam, and Cell Signaling Technology). This may be in part due its reported short life half-life [PMID: 17043353]. Since we could not consistently detect endogenous INSIG,1 we employed over expression studies using INSIG1-HA/FLAG. Our data with overexpressed INSIG1 agree with many published reports on the known role of p97-UBXD8 in regulating INSIG1 degradation [PMID: 18835813; PMID: 21115839].

3c: The results showed that the total level of SREBP1 appeared to increase upon p97 suppression, which is different from the UBXD8 KO cell. How the authors reconcile these two results?

The reviewer is correct in that the levels of total SREBP1 is elevated in p97 depleted cells relative to UBXD8 KO cells. In general, we have empirically observed that p97 depletion always produces a much stronger phenotype than depletion of specific adaptors alone. For example, p97-UFD1-NPL4 regulates CDT1 degradation at the onset of S phase. However, CDT1 is stabilized to a far greater extent by p97 depletion than by UFD1-NPL4. This occurs even though there is no redundancy with other p97 adaptors [PMID: 21981919].

3e: As mentioned above, can the authors show more MAM markers?

We have now probed our MAM fractions with other MAM markers including VAPB, SIGMA1R, and Calnexin as per reviewer suggestion. The data for is included in (Fig 1a, Supplementary Fig 1c).

Fig. 4:2b: The thinner, dotted lines in the TEM micrograph may be better for visualizing the ER and mito. In addition, can the authors confirm the TEM results with a well-established ER-PM marker MAPPER ?

We have now replaced the thick lines with thin dotted lines in TEM images as per the reviewers' suggestion. We now provide new data where we have used GFP-MAPPER to quantify the ER-PM contacts and found that upon UBXD8 or p97 knockdown and p97 inhibition (CB5083), there is an increase in ER-PM contacts. These results corroborate our TEM data showing increased ER-PM contacts in UBXD8 KO cells.

4e: The specificity of anti-SREBP1/anti-SCD1 signals in immunofluorescence staining should be carefully tested, for example, by siRNA-mediated suppression.

We have assessed the specificity of SREBP1 and SCD1 antibodies by siRNA-mediated knockdown of both SREBP1 and SCD1. The results indicate that the antibodies are specific to SREBP1 and SCD1. The data is provided in Supplementary Fig 8b.

Fig. S1:1a: Can the author show more well-established ER-mito contact tethers, for instance VAPB/PTPIP51 and VPS13A?

We thank the reviewer for this suggestion. We now provide new data where we over-express VAPB-PTPIP51 (in addition to REEP1) to quantify the ERMCS. We show ERMCS increase upon the over-expression of VAPB and PTPIP51 (Supplementary Fig 2b).

s1d: Scale bar was missing.

We apologize for the omission. We have added the scale bar.

1e: Can the authors show the dots in the bar plots?

We apologize for the omission. We have included the individual data points in the bar graph.

1f: Did p97 suppression or UBXD8 KO affects the morphology of mitochondria?

We assessed mitochondrial network morphology and area upon OA or PA supplementation using fluorescence microscopy and a published image analysis script mitochondrial network analysis (MINA) [PMID: 28314612]. Antimycin and oligomycin co-treatment was used as a positive control in wildtype cells to collapse the membrane potential and cause fragmentation (Supplementary Fig. 7d-g). Unlike antimycin-oligomycin treatment which caused significant mitochondrial fragmentation, OA and PA treatment induced very minor effects under the conditions tested and the changes were comparable between wildtype and UBXD8 KO cells (Supplementary Fig. 7d-g). Further, p97 depletion did not affect mitochondrial morphology (Supplementary Fig 8a).

Fig. S2:2b: The quantification for S2b is missing.

We have replaced the immunoblot and provided quantification from triplicate experiments (Supplementary Fig 2h).

Fig. S3 &4 Move to main figures?

The figures are now rearranged in the revised manuscript. We have revised the manuscript to now contain 8 main figures with 8 associated supplementary figures.

Fig. S5.5g: The statistical analysis is missing for the group -SCD1 inhibitor; +OA.

We apologize for the omission. We have now added the statistical analysis (Supplementary Fig 6h).

Fig. S6. 6b: off topic.

We have now removed this panel.

Reviewer#3

The AAA-ATPase p97 acts as an ubiquitin-selective unfoldase to control the abundance of several ER and mitochondria proteins involved in protein folding. P97 is recruited to the ER via its adaptors UBXD8 and UBXD2. In their manuscript Ganji et al. uses proteomics, lipidomics and TEM, as well as in vitro and in vivo cell and animal systems, to describe how inhibition of the p97-UBXD8 axis results in increased ER-mitochondria contacts. They find that this is likely due to the inability to activate SREBP1. They further showed that lipid desaturases, such as SCD1, a downstream target of SREBP1, decrease in abundance with UBXD8 KO. The authors find that decreased SCD1 expression is linked to decreased production of specific lipid species, including Oleic acid, which can be supplemented to rescue the increase in ER-mitochondria contact sites, likely due to restoration of lipid fluidity. Finally, the authors performed in vivo mouse studies to validate the clinical relevance of their findings to neurodegenerative diseases.

MajorConcerns:

1. While global membrane fluidity and lipid alterations were measured for figures 2 and 4, this does not indicate that lipid composition and fluidity are altered specifically at MAMs.

To indicate specificity to MAMs, the authors could perform lipidomics on isolated membranes, as well as use a MAM marker to measure membrane fluidity specific to those sites.

We thank the reviewer for this comment and agree that the membrane fluidity assay and whole cell lipidomics do not suggest that the lipid composition is altered at the MAMs. Hence, to evaluate lipid composition and saturation we performed lipidomics on MAM fractions isolated from HEK293T WT and UBXD8 KO cells. Out of the 195 lipids identified, 37 lipids were significantly changed in the MAM fraction of UBXD8 KO cells (Fig. 5b). Of the lipids that were increased in UBXD8 KO cells, all but one were saturated or mono-unsaturated (Fig. 5b). Furthermore, polyunsaturated lipids were significantly depleted in UBXD8 KO MAM fractions (Fig. 5b). These findings suggest that ERMCS UBXD8 KO cells have altered lipid composition and are more saturated compared to WT.

2. The authors compare the proteomes of isolated MAMs taken from WT and UBXD8 KO cells through use of TMT-MS. Through this analysis they only identify a handful of proteins being significantly altered. The authors initially note a significance cutoff of $\text{abs}(\log_2(\text{WT}/\text{KO})) > 1$; however, the only pathways noted that were not expected (either due to their roles in contacts or p97-UBXD8 pathway) are below this threshold. In following experiments, the authors identify several other proteins and processes that change in abundance in UBXD8 KO cells (SREBP1, SCD1 e.g.). Why were these proteins not identified in the TMT analysis? The knowledge added by the TMT study is not evident. Additionally, there is very little follow-up from this experiment, as most motivations for subsequent figures are from previous data.

We apologize for this error; we have now used a cut-off of $\text{Log}_2\text{FC} > \pm 0.65$ for uniformity. We agree with the reviewer that the TMT shows minimal differences in SREBP1 pathway components in our UBXD8 KO cells. While our immunoblotting studies find significant differences in SREBP and SCD1 protein levels, the same phenotype is not apparent in the TMT proteomics. This may be due to a number of reasons. The MAM purifications are not pure and there is always some organelle contamination that varies from experiment to experiment. Furthermore, ratio compression in TMT studies can repress fold changes. Even though peptide abundance is analyzed in MS3 mode, there is still some interference/compression. This can happen when (i) a low abundance protein is being analyzed and it has very few peptides (low summed signal), (ii) the peptides are not ionized well (low overall signal), (iii) the protein shares many peptides with other proteins or (iv) true interference from another more signal-dominant peptide that coelutes within the same isolation window, but has a different sequence and originates from a different protein. For instance, all the peptides quantified for SREBP1 and 2 were shared thus preventing accurate quantification. However, our proteomic studies also identified desaturases FADS1 ($\log_2\text{FC WT:UBXD8 KO} = 0.60$) and FADS2 ($\log_2\text{FC WT:UBXD8 KO} = 0.42$) to be depleted modestly in UBXD8 KO (Fig. 4b. Supplementary Table 1). The TMT study suggested altered lipid biosynthesis which necessitated the lipidomics study. We pursued the SREBP-SCD1 pathway due to the significant changes in the lipidome and yeast studies that reported a role for UBXD8 in regulating membrane lipid saturation. For clarity we now include a discussion of the ratio compression issue in the Methods section.

3. Immunofluorescence microscopy imaging should be used to validate the western blots indicating protein localization to different fractions. Additionally, the authors suggest that the observed lipid alterations with UBXD8 KO are due to the inability of SREBP1 to translocate, which would also be easily tested via immunofluorescence microscopy.

We now show enrichment of UBXD8 to ERMCS using microscopy. We imaged Cos-7 cells transiently transfected with mito-BFP, SEC61 β and mCherry-UBXD8 to observe UBXD8 localization at ERMCS. We found that relative to ER marker, mCherry-UBXD8 was enriched at ERMCS (Fig. 1c, d). We were unable to show p97 enrichment at ERMCS using microscopy as it is a very abundant protein that localizes peripherally to the ER. However, using the subcellular fractionation we show that p97 localization to ERMCS is diminished in UBXD8 KO cells (Supplementary Fig 1c-d).

To validate SREBP1 processing defects in UBXD8 KO cells, we provide new fractionation studies. We isolated nuclei and found that mature SREBP1 translocation to the nucleus was decreased in UBXD8 KO compared to WT cells (Supplementary Fig 6d).

4. The authors systematically identify important components downstream of UBXD8 throughout the paper. Inclusion of a model figure putting all the findings together into a cartoon would aid the interpretation and emphasize the pathways that are manipulated.

A model summarizing the findings has been added in the revised manuscript (Fig 8f).

Minor Concerns:

1. For figure 3d, why are all the graphs normalized to the control except for mature SREBP1?

Mature SREBP1 was calculated as a percentage of total SREBP1 (Immature + Mature SREBP1) and subsequently normalized to the loading control. Hence the graph represents the % mature SREBP1 in each sample and not the fold change.

2. In figure 3f SREBP1 and SCD1 abundances are mildly perturbed by UBXD8 KO, is this significant?

We have now provided the corresponding quantification plots and show significance based on triplicate studies (Fig 6h).

3. Addition of quantification for SCD1 abundance from the blot shown in supplementary figure 5f would make interpretation easier.

We have now provided the quantification (Supplementary Fig 6g).

4. Indicate the distance between split luciferase probes necessary for signal and how that distance compares to that found for ERMCS. Additionally, provide information on how these probes are localized to their respective organelles.

We provide new data using a fluorescent protein based split system (Split-GFP system) using to quantify ERMCS. The split-GFP constructs used in this study have been extensively validated [PMID: 29229997; PMID: 34118050]. These constructs, localize to either the ER and Mitochondrial membranes and measure close range contacts (~8-10nm). ERMCS have been reported to span a range of 5-100nm. Our results using the split-GFP assay agree with our split-luciferase assays showing increased ERMCS upon UBXD8 or p97 knockdown (Fig 3a-b).

The split-luciferase reporter system was generated by the Golden lab and has been validated to verify appropriate localization of the split reporters to the respective organelles. An increase in the luminescence of this reporter has been verified by the Golden lab and our group

using REEP1, a known ERMCS tether protein [PMID: 26201691; PMID: 28760823]. We have additionally assessed the specificity and robustness of split luciferase constructs to quantify ERMCS. We quantified the levels of partial luciferase proteins using Western blotting. We did not see any significant changes in the expression levels of split-luciferase constructs across different gene knockdown conditions, including but not limited to p97 and UBXD8 knock down by siRNA (Supplementary Fig 2c-d). Additionally, we used an established ERMCS tether complex, VAPB-PTPIP51, apart from REEP1 to quantify the ERMCS and found that the ERMCS increased upon the over-expression of VAPB and PTPIP51. Unfortunately, we do not know the precise ERMCS distance the split luciferase system reports as that was not published by the Golden lab. However, based on the sequences used and the lack of linkers between the organelle targeting sequences and the split luciferase domains, we presume that the distance is comparable to the Split GFP system and thus reports on close range contacts.

5. As validation of increased ERMCSs with UBXD8 the authors indicate that there is no significant change in mitochondria abundance; however, they do not mention whether there also are ER abundance or morphology changes.

We have now measured the ER length in the TEM micrographs (Fig 3e). We do not see any significant changes in the ER length between WT or UBXD8 KO cells.

6. The sentence in line 631-632 has a typo.

We apologize for the mistake. It has been corrected in the revised manuscript.

7. Referring to ER-mitochondria contact sites as simply “contacts” is a confusing shorthand. As an alternative, an abbreviation (e.g., ERMCS) could be considered and would be more accurate.

We now refer to ER-Mitochondrial contact sites as ERMCS throughout the revised manuscript.

8. It would be good not to modulate twice in the title.

We have removed the second instance of modulate and revised the title.

Reviewer#4

In this study, Ganji et al. utilise a variety of approaches in HEK293 cells to demonstrate that loss of the p97 AAA ATPase and its adaptor UBXD8 leads to decreased SREBP1 activation, decreased desaturase activity and membrane fluidity and this leads to increased ER-mito contact sites. Importantly, they show that this can be rescued by oleic acid supplementation or SCD1 overexpression. In addition they show that there is also increased mito-ER contact sites that can be rescued with oleic acid in cells with the IBMPFD common mutation p97 R155H, and that SCD1 and active SREBP1 levels are decreased in the brains of mouse models with conditional knockout of p97 or p97R155C/WT . Overall, these are interesting findings relating p97 to SREBP1 driven changes in lipid saturation that impact ER-mito contact sites.

My main comment is centered around the presentation and interpretation of the lipidomics data. The authors analyse a range of phospholipids, DAGs, TAGs and cholesterol esters and present the data as graphs with the data plotted as scatter charts of the log₁₀ peak

areas of the individual lipid species in the KO versus the WT or as heatmaps and dot plots showing fold change.

1) P values of lipids that change are shown in supplementary tables but it would be helpful for the reader if an indication of significance or variability in the data was also incorporated into the figures. For example, a volcano plot of KO versus WT lipid species. In addition, are all the species shown in the heat maps in figure s4 d and e significantly different or was any p value or q value cut-off applied to select which lipids to show in the heatmap?

We thank the reviewer for this comment. As per the reviewer's suggestion, we have now depicted the lipidomics data as volcano plots so that statistical significance and fold change is readily apparent. (Fig. 5 and Supplementary figure 5). We have marked select saturated or mono-unsaturated lipids in the volcano plots.

2) The authors argue that the lipid species that are most increased in the KO cells have 1 or less double bonds in the acyl chain. They then go on to demonstrate the importance of SCD and how monounsaturated or fatty acids with one double bond can rescue the phenotype. In addition, from the dot plots showing the two tails of PC, lipids containing monounsaturated fatty acids seem increased in the KO. If the aim of the lipidomic data is to highlight that there is more saturated lipids, it doesn't seem to show this (for example, the biggest fold change in PE lipids contain 46:6, 40:5, 40:6). Based on the other data in the manuscript, I don't doubt that there seems to be an increase in saturation but the lipidomic data as currently presented doesn't support this. As the data presented here shows relative abundance of individual species between cell types but not the absolute abundance of different lipid species, are the lipid species containing saturated fatty acids more abundant in the cell on a molar basis? Could the authors look at total fatty acid abundance and the ratio of C16:0/C16:1, C18:0/C18:1 to get a better quantitative indication of saturation?

As per the reviewer's suggestion, we now depicted the lipidomics data as volcano plots and labeled specific saturated or mono-unsaturated fatty acids in the volcano plots to highlight their increase in UBXD8 KO cells (Fig. 5 and Supplementary figure 5).

It is correct that the lipid measurements reported in this work represents the relative abundance of individual species between KO and WT cells. The lipidomic data were normalized by protein content. In an initial study, absolute lipid concentrations on a molar per cell basis was used to compare KO and WT cells, in addition to determining relative abundance as reported in the paper. The two comparisons were similar. Since purified lipid standards for some of the most interesting lipids altered in this study—those with saturated or monounsaturated tails—do not exist, we prefer to use the relative abundance data since we lack all the preferred standards for determining absolute lipid concentrations. While it could be reasonably presumed that the lipids with shorter tails can be used for lipids with these very long tails in the same lipid class, we felt it best to avoid presumptions. However, we recognize that each approach has caveats. As such, we further performed several quality control checks and internal controls to ensure the quantification of lipids. These include performing duplicate samples in parallel to determine variability in sample handling and mass spectrometry analyses for each independent experiment, using external standards (Avanti Polar SPLASH LIPIDOMIX, NIST Reference Material 1950), and several injection volumes (e.g., 4, 8, and 16 ul for positive mode). These are discussed in the Materials and Methods. If any lipid measurement fails a quantitative e.g., parallel duplicate samples are different—than the measurement was excluded from the study. Additionally, several 'blank' runs and 'no cell' samples were analyzed before, interspersed within, and after each sample to best exclude background contaminating peaks.

We also provide new studies wherein we performed lipidomics on MAM fractions isolated from HEK293T WT and UBXD8 KO cells. Out of the 195 lipids identified, 37 lipids were significantly changed in the MAM fraction of UBXD8 KO cells (Fig. 5b). Of the lipids that were increased in UBXD8 KO cells, all but one were saturated or mono-unsaturated (Fig. 5b). Furthermore, polyunsaturated lipids were significantly depleted in UBXD8 KO MAM fractions (Fig. 5b). These findings suggest that ERMCS UBXD8 KO cells have altered lipid composition and are more saturated compared to WT.

3) Is there a mistake in s4f as it seems to contain two identical dot plots?

We apologize for the error and appreciate the reviewer for pointing it out. As we have now reformatted all the lipidomics data, the dot plots have been removed.

4) Does UBXD8 loss impact mitochondria function such as respiration, and does oleic acid rescue this?

We have assessed mitochondrial network morphology and area using fluorescence microscopy and a published image analysis script mitochondrial network analysis (MINA) [PMID: 28314612]. Antimycin and oligomycin co-treatment was used as a positive control in wildtype cells to collapse the inner mitochondrial membrane potential and cause fragmentation (Supplementary Fig. 7d-g). Unlike antimycin-oligomycin treatment which caused significant mitochondrial fragmentation, OA and PA treatment induced very minor effects under the conditions tested and the changes were comparable between wildtype and UBXD8 KO cells (Supplementary Fig. 7d-g).

We have also performed a flow cytometry-based assay using JC1, a mitochondria-specific fluorescent dye which shows a shift in spectral emission based on the mitochondrial membrane potential. We observed that the membrane potential was comparable between WT and UBXD8 KO cells (see below). We opted include the imaging studies and analysis (described above) in the manuscript.

HEK293T WT

UBXD8 KO

Mitochondrial membrane potential dye, JC1 was used to assess the mitochondrial potential in UBXD8 KO cells using flow cytometry. The JC1 dye localizes to mitochondria and exhibit potential-dependent spectral emission. Red fluorescence at high membrane potential and Green fluorescence at lower membrane potential. The data is representative of two independent biological replicates. The percentage of cells are represented as Mean \pm SEM.

REVIEWERS' COMMENTS

Reviewer #1 (Remarks to the Author):

The revised is much improved and all my previous concerns have been addressed in this version.

Reviewer #2 (Remarks to the Author):

All my concerns have been addressed. I have just one comment on the splitGFP assays of the revised manuscript. The split-GFPs are not good biosensors for monitoring the dynamics or extent of membrane contacts because they are not reversible (once reconstituted, splitGFPs are not able to dissociate). In the field, in situ proximity ligation (PLA) or ddGFP are more acceptable for this purpose.

Reviewer #3 (Remarks to the Author):

The authors have adequately addressed all previous concerns.

Reviewer #4 (Remarks to the Author):

The authors have fully addressed my comments and the lipidomics of the MAMs fraction significantly strengthens their saturation point.

Response to Review

We thank all the reviewers for agreeing to the revisions made. The reviews are reproduced in their entirety (bold), our responses are in blue.

REVIEWERS' COMMENTS

Reviewer #1 (Remarks to the Author):

The revised is much improved and all my previous concerns have been addressed in this version.

Thank you for agreeing with the revisions made.

Reviewer #2 (Remarks to the Author):

All my concerns have been addressed. I have just one comment on the splitGFP assays of the revised manuscript. The split-GFPs are not good biosensors for monitoring the dynamics or extent of membrane contacts because they are not reversible (once reconstituted, splitGFPs are not able to dissociate). In the field, in situ proximity ligation (PLA) or ddGFP are more acceptable for this purpose.

Thank you for agreeing with the revisions made. We agree with the reviewer about the limitations of the split-GFP being irreversible once reconstituted. Hence, we have now clarified the same in the results section.

Reviewer #3 (Remarks to the Author):

The authors have adequately addressed all previous concerns.

Thank you for agreeing with the revisions made.

Reviewer #4 (Remarks to the Author):

The authors have fully addressed my comments and the lipidomics of the MAMs fraction significantly strengthens their saturation point.

Thank you for agreeing with the revisions made.